# A community-maintained standard library of population genetic models

**Jeffrey R Adrion**[1†§], **Christopher B Cole**[2†§], **Noah Dukler**[3†§], **Jared G Galloway**[1†§], **Ariella L Gladstein**[4†§], **Graham Gower**[5†§], **Christopher C Kyriazis**[6†§], **Aaron P Ragsdale**[7†§], **Georgia Tsambos**[8†§], **Franz Baumdicker**[9], **Jedidiah Carlson**[10], **Reed A Cartwright**[11], **Arun Durvasula**[12], **Ilan Gronau**[13], **Bernard Y Kim**[14], **Patrick McKenzie**[15], **Philipp W Messer**[16], **Ekaterina Noskova**[17], **Diego Ortega-Del Vecchyo**[18], **Fernando Racimo**[5], **Travis J Struck**[19], **Simon Gravel**[7‡#], **Ryan N Gutenkunst**[19‡#], **Kirk E Lohmueller**[6,12‡#], **Peter L Ralph**[1,20‡#], **Daniel R Schrider**[4‡#], **Adam Siepel**[3‡#], **Jerome Kelleher**[21‡¶*], **Andrew D Kern**[1‡¶*]

[1]Department of Biology and Institute of Ecology and Evolution, University of Oregon, Eugene, United States; [2]Weatherall Institute of Molecular Medicine, University of Oxford, Oxford, United Kingdom; [3]Simons Center for Quantitative Biology, Cold Spring Harbor Laboratory, Cold Spring Harbor, United States; [4]Department of Genetics, University of North Carolina at Chapel Hill, Chapel Hill, United States; [5]Lundbeck GeoGenetics Centre, Globe Institute, University of Copenhagen, Copenhagen, Denmark; [6]Department of Ecology and Evolutionary Biology, University of California, Los Angeles, Los Angeles, United States; [7]Department of Human Genetics, McGill University, Montreal, Canada; [8]Melbourne Integrative Genomics, School of Mathematics and Statistics, University of Melbourne, Melbourne, Australia; [9]Department of Mathematical Stochastics, University of Freiburg, Freiburg, Germany; [10]Department of Genome Sciences, University of Washington, Seattle, United States; [11]The Biodesign Institute and The School of Life Sciences, Arizona State University, Tempe, United States; [12]Department of Human Genetics, David Geffen School of Medicine, University of California, Los Angeles, Los Angeles, United States; [13]The Efi Arazi School of Computer Science, Herzliya Interdisciplinary Center, Herzliya, Israel; [14]Department of Biology, Stanford University, Stanford, United States; [15]Department of Ecology, Evolution, and Environmental Biology, Columbia University, New York, United States; [16]Department of Computational BiologyCornell University, Ithaca, United States; [17]Computer Technologies Laboratory, ITMO University, Saint Petersburg, Russian Federation; [18]International Laboratory for Human Genome Research, National Autonomous University of Mexico, Juriquilla, Mexico; [19]Departmentof Molecular and Cellular Biology, University of Arizona, Tucson, United States; [20]Department of Mathematics, University of Oregon, Eugene, United States; [21]Big Data Institute, Li Ka Shing Centre for Health Information and Discovery, University of Oxford, Oxford, United Kingdom

**\*For correspondence:**
jerome.kelleher@bdi.ox.ac.uk (JK); adkern@uoregon.edu (ADK)

[†]These authors contributed equally to this work
[‡]These authors also contributed equally to this work
[§]Co-first authors are listed alphabetically
[#]Co-senior authors are listed alphabetically
[¶]Co-corresponding authors are listed alphabetically

**Abstract** The explosion in population genomic data demands ever more complex modes of analysis, and increasingly, these analyses depend on sophisticated simulations. Recent advances in population genetic simulation have made it possible to simulate large and complex models, but specifying such models for a particular simulation engine remains a difficult and error-prone task.

Computational genetics researchers currently re-implement simulation models independently, leading to inconsistency and duplication of effort. This situation presents a major barrier to empirical researchers seeking to use simulations for power analyses of upcoming studies or sanity checks on existing genomic data. Population genetics, as a field, also lacks standard benchmarks by which new tools for inference might be measured. Here, we describe a new resource, stdpopsim, that attempts to rectify this situation. Stdpopsim is a community-driven open source project, which provides easy access to a growing catalog of published simulation models from a range of organisms and supports multiple simulation engine backends. This resource is available as a well-documented python library with a simple command-line interface. We share some examples demonstrating how stdpopsim can be used to systematically compare demographic inference methods, and we encourage a broader community of developers to contribute to this growing resource.

## Introduction

While population genetics has always used statistical methods to make inferences from data, the degree of sophistication of the questions, models, data, and computational approaches used have all increased over the past two decades. Currently, there exist a myriad of computational methods that can infer the histories of populations (*Gutenkunst et al., 2009*; *Li and Durbin, 2011*; *Excoffier et al., 2013*; *Schiffels and Durbin, 2014*; *Terhorst et al., 2017*; *Ragsdale and Gravel, 2019*), the distribution of fitness effects (*Boyko et al., 2008*; *Kim et al., 2017*; *Tataru et al., 2017*; *Fortier et al., 2019*; *Huang and Siepel, 2019*; *Vecchyo et al., 2019*), recombination rates (*McVean et al., 2004*; *Chan et al., 2012*; *Lin et al., 2013*; *Adrion et al., 2020*; *V Barroso et al., 2019*), and the extent of positive selection in genome sequence data (*Kim and Stephan, 2002*; *Eyre-Walker and Keightley, 2009*; *Alachiotis et al., 2012*; *Garud et al., 2015*; *DeGiorgio et al., 2016*; *Kern and Schrider, 2018*; *Sugden et al., 2018*). While these methods have undoubtedly increased our understanding of genetic and evolutionary processes, very little has been done to systematically benchmark the quality of these inferences or their robustness to deviations from their underlying assumptions. As large databases of population genetic variation begin to be used to inform public health procedures, the accuracy and quality of these inferences is becoming ever more important.

Assessing the accuracy of inference methods for population genetics is challenging in large part because the 'ground-truth' in question generally comes not from direct empirical observations, as the relevant historical processes can rarely be observed, but instead from simulations. Population genetic simulations are therefore critically important to the field, yet there has been no systematic attempt to establish community standards or best practices for executing them. Instead, the general modus operandi to date has been for individual groups to validate their own methods using simulations coded from scratch. Often these simulations are more useful to showcase a novel method than to rigorously compare it with competing methods. Moreover, this situation results in a great deal of duplicated effort, and contributes to decreased reproducibility and transparency across the entire field. It is also a barrier to entry to the field, because new researchers can struggle with the many steps involved in implementing a state-of-the-art population genetics simulation, including identifying appropriate demographic models from the literature, translating them into input for a simulator, and choosing appropriate values for key population genetic parameters, such as the mutation and recombination rates.

A related issue is that it has been challenging to assess the degree to which modeling assumptions and choices of data summaries can affect population genetic inferences. Standardized simulations would enable these questions to be systematically examined. Importantly, there are clear examples of different methods yielding fundamentally different conclusions. For example, Markovian coalescent methods applied to human genomes have suggested large ancient (> 100,000 years ago) ancestral population sizes and bottlenecks that have not been detected by other methods based on allele frequency spectra (see *Beichman et al., 2017*). These distinct methods differ in how they model, summarize, and optimize fit to genetic variation data, suggesting that such design choices can greatly affect the performance of the inference. Furthermore, some methods are likely to perform better than others under certain scenarios, but researchers lack principled guidelines for

selecting the best method for addressing their particular questions. The need for guidance from simulated data will only increase as researchers seek to apply population genetic methods to a growing collection of non-model taxa.

For these reasons, we have generated a standardized, community-driven resource for simulating published demographic models from a number of popular study systems. This resource, which we call `stdpopsim`, makes running realistic simulations for population genetic analysis a simple matter of choosing pre-implemented models from a community-maintained catalog. The `stdpopsim` catalog currently contains six species: humans, *Pongo abelii*, *Canis familiaris*, *Drosophila melanogaster*, *Arabidopsis thaliana*, and *Escherichia coli*. For each species, the catalog contains curated information on our current understanding of the physical organization of its genome, inferred genetic maps, population-level parameters (e.g. mutation rate and generation time estimates), and published demographic models. These models and parameters are meant to represent the field's current understanding, and we intend for this resource to evolve as new results become available, and other existing models are added to `stdpopsim` by the community. We have implemented both a command line interface and a simple Python API that can be used to simulate genomic data from a choice of organism, genetic map, chromosome, and demographic history. In this way, `stdpopsim` will lower the barrier to high-quality simulation for exploratory analyses, enable rigorous evaluation of population genetic software, and contribute to increased reliability of population genetic inferences.

The `stdpopsim` library has been developed by the `PopSim` Consortium using a distributed open source model, with strong procedures in place to continue its growth and maintain quality. Importantly, we developed rigorous quality control methods to ensure that we have correctly implemented the models as described in their original publication and provided documented methods for others to contribute new models. We invite new collaborators to join our community: those interested should visit our developer documentation at https://stdpopsim.readthedocs.io/en/latest/development.html. Below we describe the resource and give examples of how it can be used to benchmark demographic inference methods.

## Results

The `stdpopsim` library is a community-maintained collection of empirical genome data and population genetics simulation models, illustrated in *Figure 1*. The package (https://github.com/popsim-consortium/stdpopsim) centers on a catalog of genomic information and demographic models for a growing list of species (*Figure 1A*), and software resources to facilitate efficient simulations (*Figure 1B–C*). Given the genome data and simulation model descriptions defined within the library, it is straightforward to run standardized simulations across a range of organisms. `Stdpopsim` has a Python API and a user-friendly command line interface, allowing users with minimal experience direct access to state-of-the-art simulations. Simulations are output in the 'succinct tree sequence' format (*Kelleher et al., 2016*; *Kelleher et al., 2018*; *Kelleher et al., 2019*), which contains complete genealogical information about the simulated samples, is extremely compact, and can be processed efficiently using the tskit library (*Kelleher et al., 2016*; *Kelleher et al., 2018*). The tree sequence format could also be converted to other formats (e.g., VCF) by the user if desired.

### The species catalog

The central feature of `stdpopsim` is the species catalog, a systematic organization of the key quantitative data needed to simulate a given species. Data are currently available for humans, *P. abelii*, *C. familiaris*, *D. melanogaster*, *A. thaliana*, and *E. coli*. A species definition consists of two key elements. Firstly, the library defines some basic information about our current understanding of each species' genome, including information about chromosome lengths, average mutation rate estimates, and generation times. We also provide access to detailed empirical information such as inferred genetic maps, which model observed heterogeneity in recombination rate along chromosomes. Such maps are often large, so we do not distribute them directly with the software, but make them available for download in a standard format. When a simulation using such a map is requested by the user, `stdpopsim` will transparently download the map data into a local cache, where it can be quickly retrieved for subsequent simulations. In the initial version of `stdpopsim`, we support the HapMapII (*Frazer et al., 2007*) and deCODE (*Kong et al., 2010*) genetic maps for humans; the *Nater et al.,*

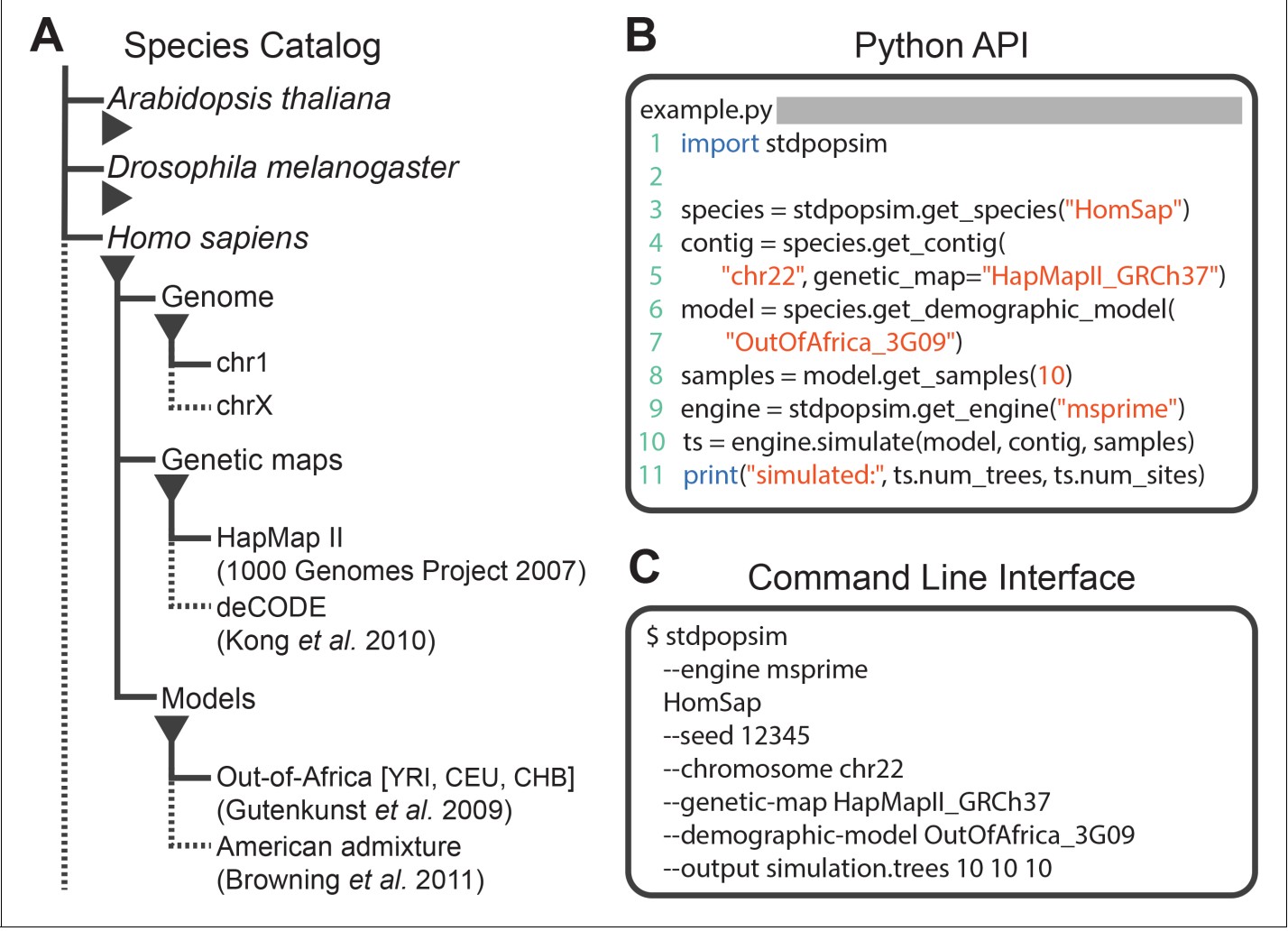

**Figure 1.** Structure of stdpopsim. (A) The hierarchical organization of the stdpopsim catalog contains all model simulation information within individual species (expanded information shown here for *H. sapiens* only). Each species is associated with a representation of the physical genome, and one or more genetic maps and demographic models. Dotted lines indicate that only a subset of these categories is shown. At right we show example code to specify and simulate models using (B) the python API or (C) the command line interface.

*2017* maps for *P. abelii*; the *Campbell et al., 2016* map for *C. familiaris*; the *Salomé et al., 2012* map for *A. thaliana*; and the *Comeron et al., 2012* map for *D. melanogaster*. Adding further maps to the library is straightforward. The second key element of a species description within stdpopsim is a set of carefully curated population genetic model descriptions from the literature, which allow simulation under specific historical scenarios that have been fit to present-day patterns of genetic variation (see the Materials and methods for a description of the community development and quality-control process for these models.)

The current demographic models in the stdpopsim catalog are shown in *Table 1*. *Homo sapiens* currently has the richest selection of population models. These include: a simplified version of the *Tennessen et al., 2012* model with only the African population specified (expansion from the ancestral population and recent growth; Africa_1T12); the three-population model of *Gutenkunst et al., 2009*, which specifies the out-of-Africa bottleneck as well as the subsequent divergence of the European and Asian populations (OutOfAfrica_3G09); the *Tennessen et al., 2012* two-population variant of the Gutenkunst et al. model, which does not include Asian populations but more explicitly models recent rapid human population growth in Europe (OutOfAfrica_2T12); the *Browning et al., 2018* admixture model for American populations, which specifies ancestral African, European, and Asian population components (AmericanAdmixture_4B11); a three-population

**Table 1.** Initial set of demographic models in the catalog and summary of computing resources needed for simulation.
For each model, we report the CPU time, maximum memory usage and the size of the output tskit file, as simulated using the msprime simulation engine (version 0.7.4). In each case, we simulate 100 samples drawn from the first population, for the shortest chromosome of that species and a constant chromosome-specific recombination rate. The times reported are for a single run on an Intel i5-7600K CPU. Computing resources required will vary widely depending on sample sizes, chromosome length, recombination rates and other factors.

| Model ID | Citation | CPU(s) | Ram(MB) | File(MB) |
|---|---|---|---|---|
| HomSap (*Homo sapiens*) | | | | |
| Africa_1T12 | *Tennessen et al., 2012* | 10.0 | 194.2 | 23.3 |
| Zigzag_1S14 | *Schiffels and Durbin, 2014* | 3.3 | 106.1 | 7.9 |
| AshkSub_7G19 | *Gladstein and Hammer, 2019* | 13.8 | 216.3 | 26.4 |
| OutOfAfrica_3G09 | *Gutenkunst et al., 2009* | 10.2 | 182.0 | 21.1 |
| OutOfAfrica_2T12 | *Tennessen et al., 2012* | 10.7 | 198.4 | 24.1 |
| AncientEurasia_9K19 | *Kamm et al., 2019* | 63.1 | 304.4 | 41.2 |
| AmericanAdmixture_4B11 | *Browning et al., 2018* | 10.6 | 188.1 | 22.3 |
| PapuansOutOfAfrica_10J19 | *Jacobs et al., 2019* | 204.5 | 524.7 | 77.8 |
| OutOfAfricaArchaicAdmixture_5R19 | *Ragsdale and Gravel, 2019* | 8.8 | 185.4 | 21.7 |
| DroMel (*Drosophila melanogaster*) | | | | |
| OutOfAfrica_2L06 | *Li and Stephan, 2006* | 252.8 | 678.0 | 106.7 |
| African3Epoch_1S16 | *Sheehan and Song, 2016* | 3.0 | 123.9 | 11.5 |
| AraTha (*Arabidopsis thaliana*) | | | | |
| African2Epoch_1H18 | *Huber et al., 2018* | 4.3 | 220.5 | 16.5 |
| African3Epoch_1H18 | *Huber et al., 2018* | 2.6 | 241.3 | 18.4 |
| PonAbe (*Pongo abelii*) | | | | |
| TwoSpecies_2L11 | *Locke et al., 2011* | 7.2 | 171.9 | 14.7 |

out-of-Africa model from *Ragsdale and Gravel, 2019*, which includes archaic admixture (`OutOfAfricaArchaicAdmixture_5R19`); a complex model of ancient Eurasian admixture from *Kamm et al., 2019* (`AncientEurasia_9K19`); and a synthetic model of oscillating population size from *Schiffels and Durbin, 2014* (`Zigzag_1S14`).

For *D. melanogaster*, we have implemented the three-epoch model estimated by *Sheehan and Song, 2016* from an African sample (`African3Epoch_1S16`), as well as the out-of-Africa divergence and associated bottleneck model of *Li and Stephan, 2006*, which jointly models African and European populations (`OutOfAfrica_2L06`). For *A. thaliana*, we implemented the model in *Durvasula et al., 2017* inferred using `MSMC`. This model includes a continuous change in population size over time, rather than pre-specified epochs of different population sizes (`SouthMiddleAtlas_1D17`). We have also implemented a two-epoch and a three-epoch model estimated from African samples of *A. thaliana* in *Huber et al., 2018* (`African2Epoch_1H18` and `African3Epoch_1H18`).

In addition to organism-specific models, `stdpopsim` also includes a generic piecewise constant size model and isolation with migration (IM) model which can be used with any genome and genetic map. Together, these models contain many features believed to affect observed patterns of polymorphism (e.g. bottlenecks, population growth, admixture) and therefore provide useful benchmarks for method development.

To guarantee reproducibility, we have standardized naming conventions for species, genetic maps, and demographic models that will enable long-term stability of unique identifiers used throughout `stdpopsim`, as described in our documentation (https://stdpopsim.readthedocs.io/en/latest/development.html#naming-conventions).

## Simulation engines

Currently, `stdpopsim` uses the `msprime` coalescent simulator (*Kelleher et al., 2016*) as the default simulation engine. Coalescent simulations, while highly efficient, are limited in their ability to model continuous geography or complex selection scenarios, such as recurrent sweeps and background selection. For these reasons, we have also implemented the forward-time simulator, `SLiM` (*Haller and Messer, 2019*; *Haller and Messer, 2019*), as an alternative backend engine to `stdpopsim`, allowing for the simulation of processes that cannot be modeled under the coalescent. However, as forward-time simulators explicitly model all individuals in a population, simulating large population sizes can be highly demanding of computational resources. One common practice used to address this challenge is to simulate a *smaller* population, but to rescale resulting times, mutation rates, recombination rates, and selection coefficients so that the intensity of mutation, recombination, and allele frequency change due to selection per unit time remains the same (see the `SLiM` manual and *Uricchio and Hernandez, 2014*). Our implementation of the `SLiM` backend allows easy use of this *rescaling* through a single 'scaling factor' argument. Such down-scaled simulations are not completely equivalent to simulating all individuals in the population, and may lead to subtle differences, especially in the presence of selection. However, since many sequence-based measures of population diversity remain nearly unchanged when rescaling in this fashion, this practice is effective for many purposes and widely employed.

We validated our implementation of the `SLiM` engine by comparing estimates of several population genetic summary statistics for neutral simulations generated by both `SLiM` and msprime. Examples of this validation for the `AncientEurasia_9K19` model (*Kamm et al., 2019*) are shown in *Appendix 1—figure 1* and *Appendix 1—figure 2*. For this model, down-scaling factors of up to 10 produce patterns of both diversity and linkage disequilibrium that are indistinguishable from those observed under the coalescent (i.e. msprime). Scaling down by a factor of 50 does appear to modify the distribution of these sequence statistics. Interestingly, the apparent difference between distributions is somewhat larger when simulating using a uniform recombination rate (*Appendix 1—figure 2*), likely due to the lower variation in the values of these statistics. Importantly, both comparisons validate the equivalence of SLiM and msprime when no down-scaling is applied. The results are also optimistic about the rescaling strategy to reduce computational burden, but the possible effects are not well-understood, so results relying on rescaled simulations should be carefully validated.

## Documentation and reproducibility

The `stdpopsim` command-line interface, by default, outputs citation information for the models, genetic maps, and simulation engines used in any particular run. We hope that this feature will encourage users to appropriately acknowledge the resources used in published work, and encourage authors publishing demographic models to contribute to our ongoing community-driven development process. Together with the `stdpopsim` version number and the long-term stable identifiers for population models and genetic maps, this citation information will result in well-documented and reproducible simulation workflows. The individual tree sequence files produced by `stdpopsim` also contain complete provenance information including the command line arguments, operating system environment and versions of key libraries used.

## Use case: comparing methods of demographic inference

As an example of the utility of `stdpopsim`, we demonstrate how it can be easily used to perform a fair comparison of popular demographic inference methods. Although we present comparison of results from several methods, our aim at this stage is not to provide an exhaustive evaluation or ranking of these methods. Our hope is instead to demonstrate how `stdpopsim` will facilitate more detailed future explorations of the strengths and weaknesses of the numerous inference methods that are available to the population genetics community (see Discussion).

We start by comparing popular methods for estimating population size histories of single populations and subsequently show simple examples of multi-population inference. To reproducibly evaluate and compare the performance of inference methods, we developed workflows using `snakemake` (*Köster and Rahmann, 2012*), available from https://github.com/popsim-consortium/analysis, that allow efficient computing in multicore or cluster environments. Our workflow generates $R$ replicates of $C$ chromosomes, producing $n$ population samples in each of a total of $R \times C$

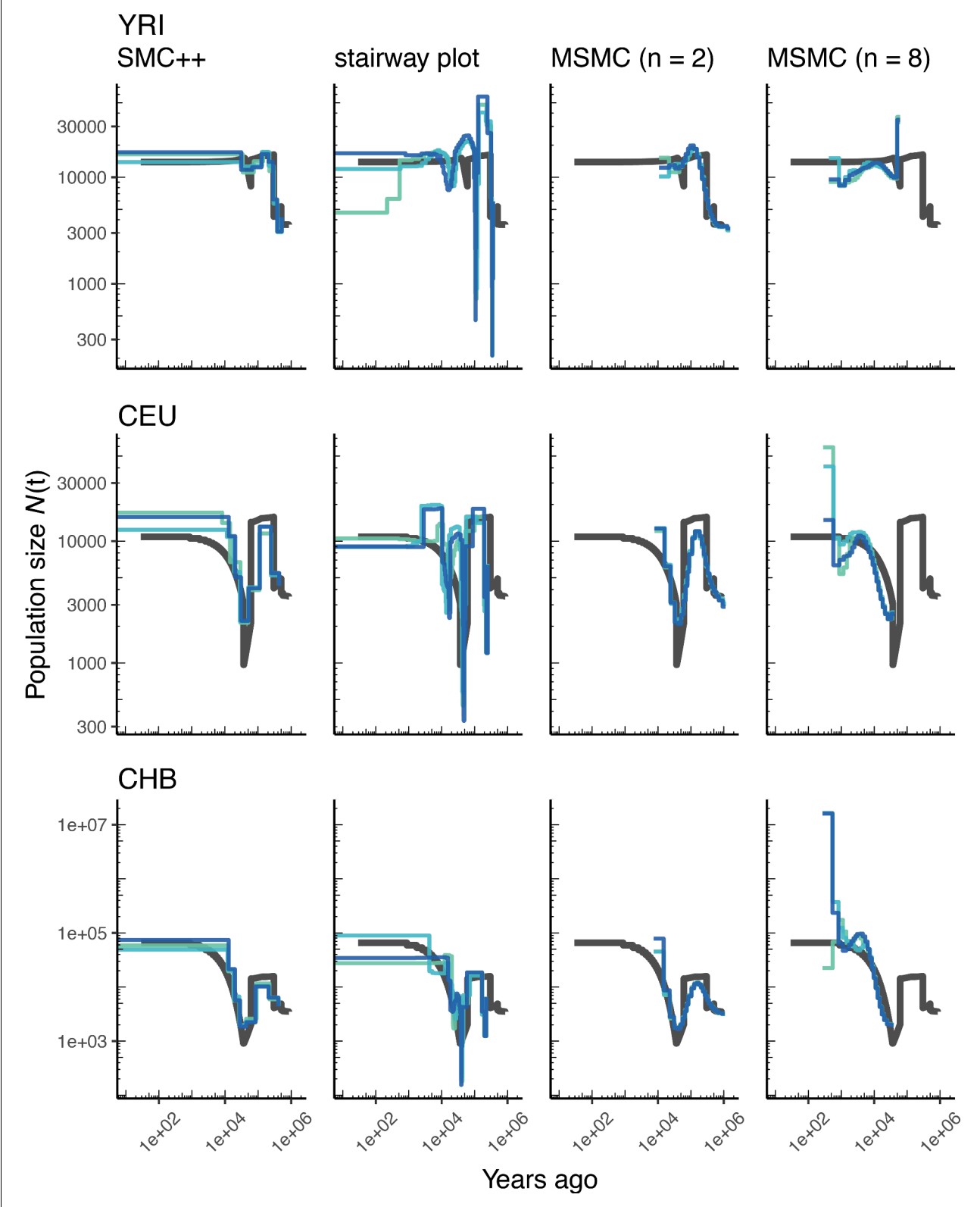

**Figure 2.** Comparing estimates of $N(t)$ in humans. Here we show estimates of population size over time ($N(t)$) inferred using four different methods: smc++, stairway plot, and MSMC with $n = 2$ and $n = 8$ samples. Data were generated by simulating replicate human genomes under the `OutOfAfricaArchaicAdmixture_5R19` model (**Ragsdale and Gravel, 2019**) and using the `HapMapII_GRCh37` genetic map (**Frazer et al., 2007**). From top to bottom, we show estimates for each of the three populations in the model (YRI, CEU, and CHB). In shades of blue we show the estimated

*Figure 2 continued on next page*

*Figure 2 continued*

$N(t)$ trajectories for each of three replicates. As a proxy for the 'truth', in black we show inverse coalescence rates as calculated from the demographic model used for simulation (see text).

simulations for each demographic model. After simulation, the workflow prepares input files for each inference method by grouping all $n \times R \times C$ simulated chromosomes into a single file. Each file is then converted into an input file appropriate for each inference method (such that all inference methods run on the same simulation replicates). Each of the inference programs are then run in parallel, and finally, estimates of population size history from each program are plotted.

## Single-population demographic models

For single-population demographic models, we compared `MSMC` (*Schiffels and Durbin, 2014*), SMC ++ (*Terhorst et al., 2017*), and `stairway plot` (*Liu and Fu, 2015*) on simulated genomes sampled from a single population, under several of the demographic models described above. However, these experiments raise the question of what to use as the 'true' population sizes in the case of multi-population models with migration. In particular, a simple single-population model that is fit to data simulated under a multi-population model, is not expected to recover the actual simulated population sizes because of model misspecification. Instead, we argue that the best one may expect in such a scenario is to infer a model that accurately reflects the coalescence time distribution of the simulated model. Under a multi-population model, the coalescence time distribution is influenced by migration between the target population and populations not analyzed in inference, as well as by the ancestral effective population sizes. The inverse coalescence rate is commonly interpreted as the effective population size, since these are equal in a single-population model with random mating. We thus analytically computed inverse coalescence rates in `msprime` for each simulated model, and used them as benchmarks for the 'true' effective population sizes. See the Appendix for a precise definition and description of the inverse coalescence rate computation.

*Figure 2* presents the results from simulations under `OutOfAfricaArchaicAdmixture_5R19`, a model of human migration out of Africa that includes archaic admixture (*Ragsdale and Gravel, 2019*), along with an empirical genetic map.In each column of this figure we show the inferred population size history (denoted $N(t)$) from samples taken from each of the three extant populations in the model. In each row we show comparisons among the methods (including two sample sizes for MSMC). Blue lines show estimates from each of three replicate whole genome simulations, and black lines indicate the 'true' values depicted by the inverse coalescence rates (although in this specific model the inverse coalescence rates are very close to the simulated population sizes; *Appendix 1—figure 3*). While there is variation in accuracy among methods, populations, and individual replicates, the methods generally produce a good estimate of the true effective population sizes of the simulations, with inferred values mostly within a factor of two of the truth, and most methods inferring a bottleneck at approximately the correct time.

Using `stdpopsim`, we can readily compare performance on this benchmark to that based on a different model of human history. In *Appendix 1—figure 4*, we show estimates of $N(t)$ from simulations using the same physical and genetic maps, but from the `OutOfAfrica_3G09` demographic model that does not include archaic admixture. Again we see that each of the methods is capturing relevant parts of the population history, although the accuracy varies across time. In comparing inferences between the models it is interesting to note that $N(t)$ estimates for the CHB and CEU simulated populations are generally better across methods than estimates from the YRI simulated population.

We can also see how well methods might do at recovering the population history of a constant-sized population, with human genome architecture and genetic map. We show results of such an experiment in *Appendix 1—figure 5*. All methods recover population size within a factor of two of the simulated values; however, SMC-based methods tend to infer sinusoidal patterns of population size even though no such change is present.

As most method development for population genetics has been focused on human data, it is important to ask how such methods might perform in non-human genomes. *Figure 3* shows parameter estimates from the African3Epoch_1S16 model, originally estimated from an African sample of *D.*

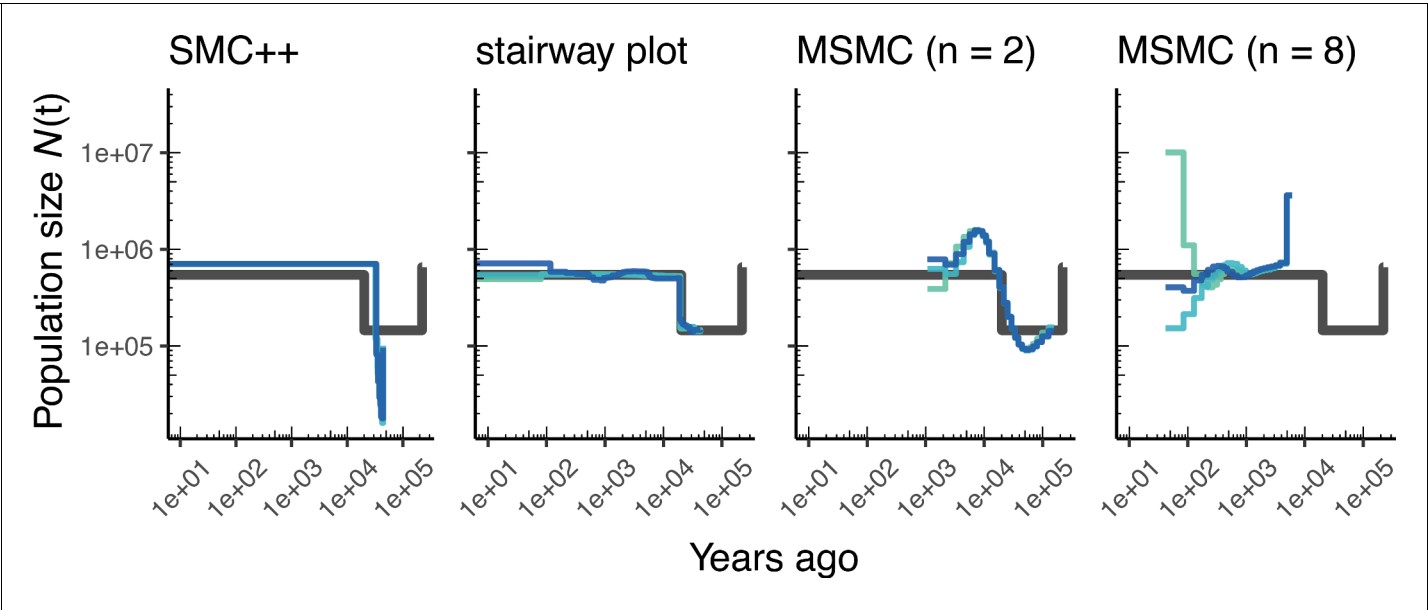

**Figure 3.** Comparing estimates of $N(t)$ in *Drosophila*. Population size over time ($N(t)$) estimated from an African population sample. Data were generated by simulating replicate *D. melanogaster* genomes under the African3Epoch_1S16 model (*Sheehan and Song, 2016*) with the genetic map of *Comeron et al., 2012*. In shades of blue we show the estimated $N(t)$ trajectories for each replicate. As a proxy for the 'truth', in black we show inverse coalescence rates as calculated from the demographic model used for simulation (see text).

melanogaster (*Sheehan and Song, 2016*), and *Appendix 1—figure 6* shows estimates from simulations of *A. thaliana* under the African2Epoch_1H18 model originally inferred by *Huber et al., 2018*. In both cases, as with humans, we use stdpopsim to simulate replicate genomes using an empirically-derived genetic map, and try to infer back parameters of the simulation model. Accuracy is mixed among methods when doing inference on simulated data from these *D. melanogaster* and *A. thaliana* models, and generally worse than what we observe for simulations of the human genome.

## Multi-population demographic models

As `stdpopsim` implements multi-population demographic models, we also explored parameter estimation of population divergence parameters. In particular, we simulated data under multi-population models for humans and *D. melanogaster* and then inferred parameters using ∂a∂i, `fastsimcoal2`, and `smc++`. For simplicity, we conducted inference in ∂a∂i and `fastsimcoal2` by fitting an isolation with migration (IM) model with constant population sizes and bi-directional migration (*Hey and Nielsen, 2004*). Our motivation for fitting this simple IM model was to mimic the typical approach of two population inference on empirical data, where the user is not aware of the 'true' underlying demography and the inference model is often misspecified. For human models with more than two populations (e.g. *Gutenkunst et al., 2009*) this limitation means that users are inferring parameters for a model that does not match the model from which the data were generated (*Figure 4A and B*). However, since the model used for inference also allows gene flow between populations, we directly compare estimated effective population sizes to the values used in simulations (black line in *Figure 4C*) and not the inverse coalescence rates.

In *Figure 4C*, we show estimates of population sizes and divergence time, for each of the inference methods, using samples drawn from African and European populations simulated under the `OutOfAfrica_3G09` model. Our results highlight many of the strengths and weaknesses of the different methods. For instance, the SFS-based approaches with simple IM models do not capture recent exponential growth in the CEU population, but do consistently recover the simulated YRI population size history. Moreover, these approaches allow migration rates to be estimated (*Appendix 1—figure 7*), and lead to more accurate inferences of divergence times. However, these migration rate estimates are somewhat biased. In contrast, `smc++` is much better at capturing the recent

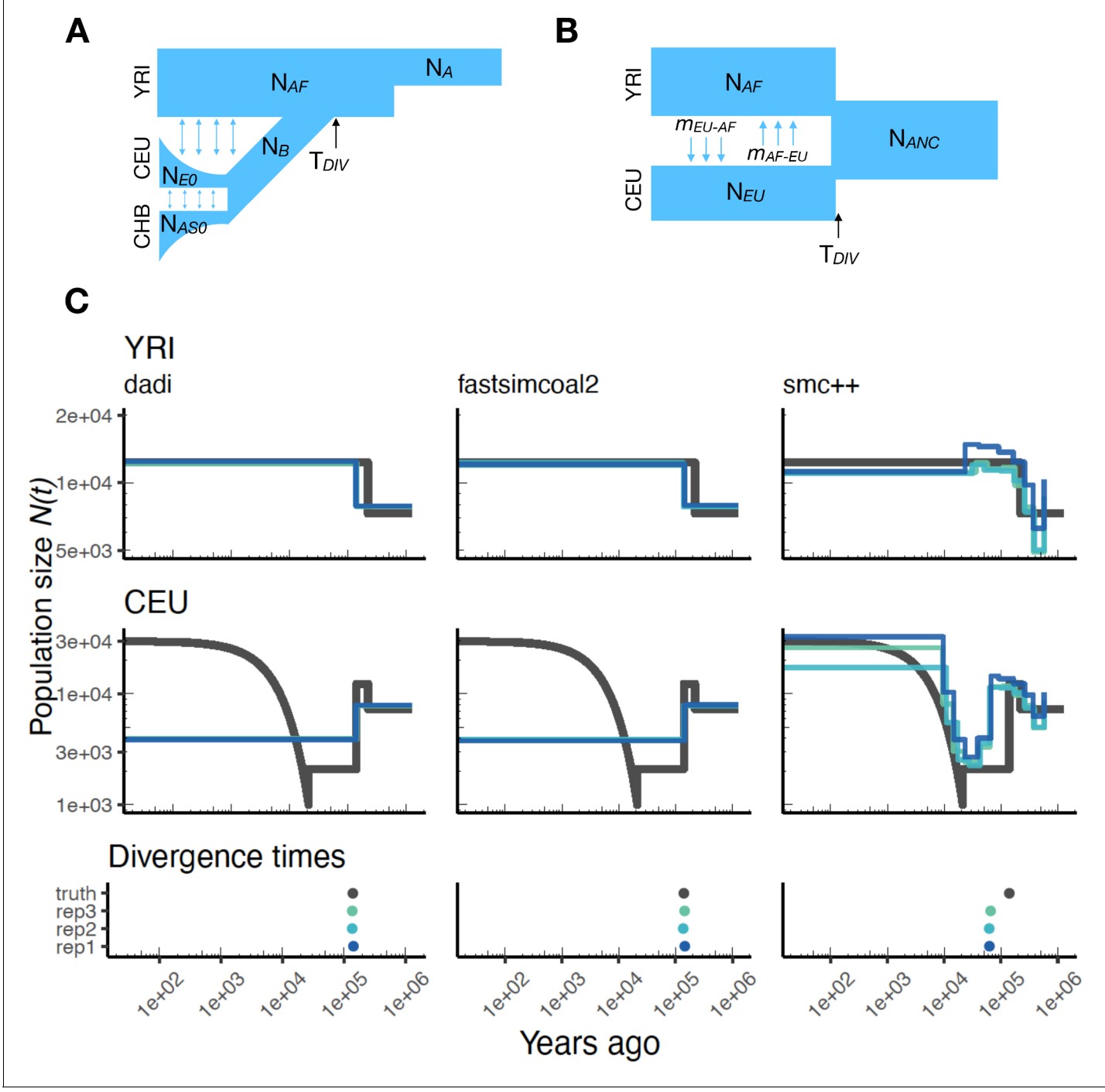

**Figure 4.** Parameters estimated using a multi-population human model. Here we show estimates of $N(t)$ inferred using ∂a∂i, fastsimcoal2, and smc++. (**A**) Data were generated by simulating replicate human genomes under the OutOfAfrica_3G09 model and using the HapMapII_GRCh37 genetic map inferred in *Frazer et al., 2007*. (**B**) For ∂a∂i and fastsimcoal2 we show parameters inferred by fitting the depicted IM model, which includes population sizes, migration rates, and a split time between CEU and YRI samples. (**C**) Population size estimates for each population (rows) from ∂a∂i, fastsimcoal2, and smc++ (columns). In shades of blue we show $N(t)$ trajectories estimated from each simulation, and in black simulated population sizes for the respective population. The population split time, $T_{DIV}$, is shown at the bottom (simulated value in black and inferred values in blue), with a common x-axis to the population size panels.

exponential growth in the CEU population, though it consistently underestimates divergence times because it assumes no migration between populations (*Figure 4C*).

Again, we can extend this analysis to other taxa and examine the performance of these methods for a two-population model of *D. melanogaster*. *Appendix 1—figure 8* shows inference results using data simulated under the `OutOfAfrica_2L06` model. This model includes an ancestral population in Africa from which a European population splits off following a bottleneck, with no post-divergence gene flow between the African and European population (*Appendix 1—figure 8A*). Here again, we find that ∂a∂i and `fastsimcoal2` infer more consistent histories, but they do not detect the brief bottleneck in Europe, due to the inference model not allowing for population size changes after the population split. In addition, ∂a∂i and `fastsimcoal2` both do reasonably well at correctly inferring the absence of migration (*Appendix 1—figure 9*). In contrast, the inferred demographic parameters from `smc++` are more noisy, though in some cases better capture the short bottleneck in the European population.

Although these results do not represent an exhaustive benchmarking, we have begun to highlight some of the strengths and weaknesses of these methods. Future work should build on these results and undertake more in-depth comparisons under a wider range of simulated demographic models.

## Discussion

Here, we have described the first major product from the `PopSim` Consortium: the `stdpopsim` library. We have founded the Consortium with a number of specific goals in mind: standardization of simulation within the population genetics community, increased reproducibility and ease of use of complex simulations, community-based development and decision making guiding best practices in population genetics, and benchmarking of inference methods.

The `stdpopsim` library allows for rigorous standardization of complex population genetic simulations. Population genetics, as a field, has yet to coalesce around a set of standards for the crucial task of method evaluation, which in our discipline hinges on simulation. In contrast, other fields such as structural biology (*Moult et al., 1995*) and machine learning *Russakovsky et al., 2015* have a long track record of standardized method testing. We hope that our efforts represent the beginning of what will prove to be an equally longstanding and valuable tradition in population genetics.

Besides being a resource for developers of computational methods, we aim for `stdpopsim` to be a resource for empirical researchers using genomic data. For instance, `stdpopsim` could be used in power analyses to determine adequate sample sizes, or in sanity checks to see if observed data (e.g. levels of divergence or the allele frequency spectrum) are roughly consistent with the hypothesized scenario. Currently, many studies would benefit from such simulation-based checks. However, there are major barriers to implementation, since individual research groups must reimplement complex, previously published demographic models, a task made especially daunting by additional layers of realism (e.g. recombination maps).

### Benchmarking population size inference

We have illustrated in this paper how `stdpopsim` can be used for direct comparisons of inferential methods on a common set of simulations. Our benchmarking comparisons have been limited, but nevertheless reveal some informative features. For example, at the task of estimating population size histories for simulated human populations, we find that the sequence-based methods (`MSMC` and `smc++`) perform somewhat better overall—at least for moderate times in the past—than the site frequency spectrum-based method (`stairway plot`), which tends to over-estimate the sizes of oscillations (*Figure 2* and *Appendix 1—figure 4*). In contrast, stairway plot outperforms the sequence-based methods on simulations of *D. melanogaster* or *A. thaliana* populations, in which linkage disequilibrium is reduced (*Figure 3* and *Appendix 1—figure 6*). In simulations of two human populations (*Figure 4*), ∂a∂i and fastsimcoal2 do reasonably well at reconstructing the simulated YRI history and estimating divergence times, but struggle with the more complex simulated CEU history, in large part because the methods assume constant population sizes. On the other hand, smc++ does not have the same restrictions on its inferred history, and as a result does much better with the CEU history but tends to underestimate divergence times due to the assumption of no migration. The results for the two-population *D. melanogaster* model (*Appendix 1—figure 8*) are generally similar. In these comparisons, fastsimcoal2 and ∂a∂i perform almost identically, which is expected because

they fit the same models to the same summaries of the data, differing only in how they calculate model expectations and optimize parameters.

All methods for inferring demographic history have strengths and weaknesses (as recently reviewed by *Beichman et al., 2018*). We compared inferences from simulated whole genome data, but many factors affect choice of methodology. Markovian coalescent methods (MSMC and smc++) require long contiguous stretches of sequence data. In contrast, frequency spectrum methods (stairway plot, $\partial a \partial i$, and `fastsimcoal2`) can use reduced-representation sequencing data, such as RAD-seq (*Andrews et al., 2016*). $\partial a \partial i$ and `fastsimcoal2` require a pre-specified parametric model, unlike `MSMC`, `smc++`, and stairway plot. Using a parametric approach yields less noisy results, but a model that is too simple may not capture important demographic events (*Figure 4* and *Appendix 1—figure 8*), and other forms of model misspecification may also produce undesirable behavior. From a software engineering perspective, methods also differ in their ease of installation and use. We hope our workflows will assist in the application of all the methods we have considered.

Altogether, these preliminary experiments highlight the utility of `stdpopsim` for comparing a variety of inference methods on the same footing, under a variety of different demographic models. In addition, the ability of `stdpopsim` to generate data with and without significant features, such as a genetic map or population-size changes (e.g., *Appendix 1—figure 5*), allows investigation of the failure modes of popular methods. Moreover the comparison of methods across the various genome organizations, genetic maps, and demographic histories of different organisms, provides valuable information about how methods might perform on non-human systems. Finally, comparison of results across methods or simulation runs provides an estimate of inference uncertainty, analogous to parametric bootstrapping, especially when different methods are vulnerable to model misspecification in different ways.

## Next steps

`Stdpopsim` is intended to be a fully open, community-developed project. Our implementations of genome representations and genetic maps for the some of the most common study systems in computational genetics—humans, *Drosophila*, and *Arabidopsis* (among others)—are only intended to be a starting point for future development. Researchers are invited to contribute to the resource by adding their organisms and models of choice. The `stdpopsim` resource is accompanied by clearly documented standard operating procedures that are intended to minimize barriers to entry for new developers. In this way, we expect the resource to expand and adapt to meet the evolving needs of the population genomics community.

One of our goals is to engage research communities studying other taxa, so as to expand the resource to many more species. Although we have included demographic models and recombination maps, there are many biological processes that we do not model. Some of the additions that we are enthusiastic to add are: selection (including distributions of fitness effects, maps of functional elements, both single and recurrent hitchhiking events, and selection on polygenic traits), gene conversion, mutation models (rate heterogeneity), more realistic demography (overlapping generations, separate sexes, mortality/fecundity schedules), geographic population structure, and downstream aspects of data quality (genotyping and mapping error). Moreover, an in-depth investigation into the effects of population-size rescaling under many of the above scenarios is warranted, given our preliminary findings using neutral simulations (*Appendix 1—figures 1* and *2*). Some other important processes are more challenging to model with current simulation software, such as structural variation, changing recombination maps over time, transposable elements, and context-dependent mutation.

We wish to emphasize that although the included demographic histories are some of the most widely used models for our current set of species, we anticipate the set of available models to expand as new methods and new modeling frameworks are developed. For instance, the current models all describe a small set of discrete, randomly mating populations, which are likely good approximations for deep-time population history, but may be less useful for methods describing dynamics of contemporary populations. `Stdpopsim's` framework is sufficiently general that more realistic population models will be easily incorporated, as they are published. Additional aspects of the framework, such as genome builds, will also continue to change as improvements are made to our understanding of genome structure.

## Materials and methods

### Model quality control

As a consortium we have agreed to a standardized procedure for model inclusion into `stdpopsim` that allows for rigorous quality control. Imagine Developer A wants to introduce a new model into `stdpopsim`. Developer A implements the demographic model for the relevant organism along with clear documentation of the model parameters and populations. This model is submitted as a 'pull request', where it is evaluated by a reviewer and then included as 'preliminary', but is not linked to the online documentation nor the command line interface. Developer A submits a quality control (QC) issue, after which a second developer, Developer B (perhaps found by requesting review from the broader Consortium), then independently reimplements the model from the relevant primary sources and adds an automatic unit test for equality between the QC implementation and the preliminary production model. If the two implementations are equivalent, the original model is included in `stdpopsim`. If not, we move to an arbitration process whereby A and B first try to work out the details of what went wrong. If that fails, the original authors of the published model must be contacted to resolve ambiguities. Further details of our QC process can be found in our developer documentation (https://stdpopsim.readthedocs.io/en/latest/development.html).

The possibility for error and the importance of careful qualty control was illustrated very clearly during our own development process: while carrying out the final revisions of this paper, we noticed that the `OutOfAfrica_3G09` model (*Gutenkunst et al., 2009*) had not gone through our QC process. The subsequent QC revealed that our implementation was in fact slightly wrong—migration rates had not been set to zero to the European population in the most ancient time period when there should have only been a single population. This error was propagated from the msprime documentation, where the model was presented as an illustrative example. A number of studies have been published using copies of this erroneous example code.

### Workflow for analysis of simulated data

To demonstrate the utility of `stdpopsim` we created `Snakemake` workflows (*Köster and Rahmann, 2012*) that perform demographic inference on tree sequence output from our package using a few common software packages (see *Appendix 1—figure 10* for an example workflow). Our choice of `Snakemake` allows complete reproducibility of the analyses shown, and all code is available from https://github.com/popsim-consortium/analysis.

We performed two types of demographic inference. Our first task was to infer effective population size over time (denoted $N(t)$). This was done using three software packages: `stairway plot`, which uses site frequency spectrum information only (*Liu and Fu, 2015*); MSMC (*Schiffels and Durbin, 2014*), which is based on the sequentially Markovian coalescent (SMC), run with two different sample sizes ($n = 2, 8$); and smc++ (*Terhorst et al., 2017*), which combines information from the site frequency spectrum with recombination information as in SMC-based methods. No attempt was made at trying to optimize the analysis from any particular software package, as our goal was not to benchmark performance of methods but instead show how such benchmarking could be easily done using the `stdpopsim` resource. In this spirit, we ran each software package as near to default parameters as possible. For `stairway plot`, we set the parameters `numRuns = 1` and `dimFactor = 5000`. For smc++ we used the 'estimate' run mode to infer $N(t)$ with all other parameters set to their default values. For MSMC, we used the `-fixedRecombination` option and used the default number of iterations.

For the single-population task, we ran human (`HomSap`) simulations using a variety of models (see *Table 1*): `OutOfAfricaArchaicAdmixture_5R19`, `OutOfAfrica_3G09`, and a constant-sized generic model. Each simulation used the `HapmapII_GRCh37` genetic map. For *D. melanogaster* we estimated $N(t)$ from an African sample simulated under the DroMel, `African3Epoch_1S16` model using the `Comeron2012_dm6` map. Finally, we ran simulations of *A. thaliana* genomes using the AraTha `African2Epoch_1H18` model under the `Salome2012_TAIR7` map. For each model, three replicate whole genomes were simulated and the population size estimated from those data. In all cases, we set the sample size of the focal population to $N = 50$ chromosomes.

Following simulation, low-recombination portions of chromosomes were masked from the analysis in a manner that reflects the 'accessible' subset of sites used in empirical population genomic studies

(e.g. *Danecek et al., 2011*; *Langley et al., 2012*). Specifically we masked all regions of 1 cM or greater in the lowest 5th percentile of the empirical distribution of recombination, regions which are nearly uniformly absent for empirical analysis. This approach to masking was chosen to prevent marginal trees with low or no recombination from biasing the comparisons of demographic inference methods. It should be noted that masking is not implemented within `stdpopsim` proper; tree sequences generated by `stdpopsim` are always raw and unmasked. This allows users the flexibility to implement masking approaches that are specific to their needs for downstream analysis.

Our second task was to explore inference with two-population models using some of the multi-population demographic models implemented in `stdpopsim`. For HomSap, we used the `OutOfA-frica_3G09` model with the `HapmapII_GRCh37` genetic map, and for DroMel we used the OutO-fAfrica_2L06 model with the `Comeron2012_dm6` map. The HomSap model is a three population model (Africa, Europe, and Asia) including post-divergence migration and exponential growth (*Figure 4C*), whereas the DroMel model is a two population model (Africa and Europe) with no post-divergence migration and constant population sizes (*Appendix 1—figure 8*).

To conduct inference on these models, we applied three commonly used methods: ∂a∂i(*Gutenkunst et al., 2009*), `fastsimcoal2` (*Excoffier et al., 2013*), and `smc++` (*Terhorst et al., 2017*). As above, these methods were used generally with default settings and we did not attempt to optimize their performance or fit parameter-rich demographic models.

For both ∂a∂i and `fastsimcoal2`, we fit a two population isolation-with-migration (IM) model with constant population sizes. This IM model contains six parameters: the ancestral population size, the sizes of each population after the split, the divergence time, and two migration rate parameters. Importantly, this meant that for both species, the fitted model did not match the simulated model (*Figure 4* and *Appendix 1—figure 8*). In the HomSap case, we therefore performed inference solely on the Africa and Europe populations, meaning that the Asia population functioned as a 'ghost' population that was ignored by our inference. To validate our inference approach, we also conducted inference on a generic IM model that was identical to the model used for inference (*Appendix 1—figure 11*).

From HomSap simulations, we took 20 whole genome samples each from the Europe and Africa populations from each replicate. Runtimes of DroMel simulations were prohibitively slow when simulating whole genomes with the `Comeron2012_dm6` map due to large effective population sizes leading to high effective recombination rates. For this reason, we present only data from 50 samples of a 3 MB region of chromosome 2R from simulations under `OutOfAfrica_2L06`. For the generic IM simulations, we used the HomSap genome along with the `HapmapII_GRCh37` genetic map and sampled 20 individuals from each population.

Following simulation, we output tree sequences and masked low-recombination regions using the same approach described for the single population workflow above. We converted tree sequences into a two-dimensional site frequency spectrum for all chromosomes in the appropriate format for ∂a∂i and `fastsimcoal2`. For each simulation replicate, we performed 10 runs of ∂a∂i and `fastsimcoal2`, checking to ensure that each method reached convergence.

Detailed settings for ∂a∂i and `fastsimcoal2` can be found in the Snakefile on our git repository (https://github.com/popsim-consortium/analysis). Estimates from the highest log-likelihood (out of 10 runs) for each simulation replicate are shown in *Figure 4C* and *Appendix 1—figure 8C*.

For `smc++`, we converted the tree sequences into VCF format and performed inference with default settings. Importantly, `smc++` assumes no migration post-divergence, deviating from the simulated model. However, because `smc++` allows for continuous population size changes, it is better equipped to capture many of the more complex aspects of the simulated demographic models (e.g. exponential growth).

To visualize our results, we plotted the inferred population size trajectories for each simulation replicate alongside the simulated population sizes (*Figure 4C* and *Appendix 1—figure 8C*). Here, unlike the single-population workflow, we compare our inferred population sizes only to the simulated population sizes and not the inverse coalescence rates.

## Resource availability

The stdpopsim package is available for download on the Python Package Index: https://pypi.org/project/stdpopsim/. Documentation for the project can be found here: https://stdpopsim.readthe-docs.io/en/latest/.

## Acknowledgements

We thank the Probabilistic Modeling in Genomics conference organizers for making this collaboration possible, and the Simons Center for Quantitative Biology at Cold Spring Harbor Laboratory for sponsoring the first workshop. Early on in the project we were encouraged by many people including Patrick Phillips, Richard Durbin, Dmitri Petrov, and Sohini Ramachandran. In addition, we thank NESCENT and Matt Hahn, Victoria Sork, and Michael Whitlock for organizing a 2014 catalysis meeting in which many of the goals of this effort were first laid out. CCK and KEL were funded under NIH Award R35GM119856. JRA and ADK were funded under NIH Award R01GM117241. TJS and RNG were funded under NIH Award R01GM127348. ALG and DRS were funded under NIH award R00HG008696. ND and AS were supported in part by NIH Awards R01HG010346 and R35GM127070. FR and GG were supported by a Villum Young Investigator award (project no. 00025300). DODV is funded by a UC MEXUS-CONACYT Collaborative Grant and a DGAPA-PAPIIT grant (PAPIIT-IA200620). JK is supported by the Robertson Foundation.

# Additional information

### Competing interests

Philipp W Messer: Reviewing editor, *eLife*. The other authors declare that no competing interests exist.

### Funding

| Funder | Grant reference number | Author |
|---|---|---|
| National Institute of General Medical Sciences | R35GM119856 | Christopher C Kyriazis Kirk E Lohmueller |
| National Institute of General Medical Sciences | R01GM117241 | Jeffrey R Adrion Andrew D Kern |
| National Institute of General Medical Sciences | R01GM127348 | Travis J Struck Ryan N Gutenkunst |
| National Institute of General Medical Sciences | R00HG008696 | Ariella L Gladstein Daniel R Schrider |
| National Institute of General Medical Sciences | R35GM127070 | Noah Dukler Adam Siepel |
| National Human Genome Research Institute | R01HG010346 | Noah Dukler Adam Siepel |
| Villum Fonden | 00025300 | Graham Gower Fernando Racimo |
| University of California Institute for Mexico and the United States | UC MEXUS-CONACYT Collaborative Grant | Diego Ortega Del Vecchyo |
| Consejo Nacional de Ciencia y Tecnología | UC MEXUS-CONACYT Collaborative Grant | Diego Ortega Del Vecchyo |
| Dirección General de Asuntos del Personal Académico, Universidad Nacional Autónoma de México | PAPIIT-IA200620 | Diego Ortega Del Vecchyo |
| Robertson Foundation | | Jerome Kelleher |

The funders had no role in study design, data collection and interpretation, or the decision to submit the work for publication.

### Author contributions

Jeffrey R Adrion, Christopher B Cole, Noah Dukler, Jared G Galloway, Ariella L Gladstein, Graham Gower, Christopher C Kyriazis, Aaron P Ragsdale, Georgia Tsambos, Major contribution to stdpopsim, documentation or analysis; Franz Baumdicker, Jedidiah Carlson, Reed A Cartwright, Arun

Durvasula, Ilan Gronau, Bernard Y Kim, Patrick McKenzie, Philipp W Messer, Ekaterina Noskova, Diego Ortega-Del Vecchyo, Fernando Racimo, Travis J Struck, Contribution to software/significant community contribution; Simon Gravel, Ryan N Gutenkunst, Kirk E Lohmueller, Peter L Ralph, Daniel R Schrider, Adam Siepel, Jerome Kelleher, Andrew D Kern, Conceptualization, Methodology, Software, Validation, Formal Analysis, Resources, Data Curation, Writing - Original Draft Preparation, Writing - Review & Editing, Supervision, Project Administration

**Author ORCIDs**
Jeffrey R Adrion ⬤ https://orcid.org/0000-0003-1021-6000
Christopher B Cole ⬤ https://orcid.org/0000-0002-6733-633X
Noah Dukler ⬤ https://orcid.org/0000-0002-8739-8052
Ariella L Gladstein ⬤ https://orcid.org/0000-0001-7735-2336
Graham Gower ⬤ https://orcid.org/0000-0002-6197-3872
Christopher C Kyriazis ⬤ https://orcid.org/0000-0002-8771-3681
Aaron P Ragsdale ⬤ https://orcid.org/0000-0003-0715-3432
Georgia Tsambos ⬤ https://orcid.org/0000-0001-7001-2275
Reed A Cartwright ⬤ https://orcid.org/0000-0002-0837-9380
Arun Durvasula ⬤ http://orcid.org/0000-0003-0631-3238
Ilan Gronau ⬤ https://orcid.org/0000-0001-8536-4062
Bernard Y Kim ⬤ https://orcid.org/0000-0002-5025-1292
Patrick McKenzie ⬤ https://orcid.org/0000-0002-8983-6060
Philipp W Messer ⬤ http://orcid.org/0000-0001-8453-9377
Ekaterina Noskova ⬤ http://orcid.org/0000-0003-1168-0497
Diego Ortega-Del Vecchyo ⬤ https://orcid.org/0000-0003-4054-3766
Fernando Racimo ⬤ https://orcid.org/0000-0002-5025-2607
Simon Gravel ⬤ https://orcid.org/0000-0002-9183-964X
Ryan N Gutenkunst ⬤ https://orcid.org/0000-0002-8659-0579
Kirk E Lohmueller ⬤ https://orcid.org/0000-0002-3874-369X
Peter L Ralph ⬤ https://orcid.org/0000-0002-9459-6866
Daniel R Schrider ⬤ https://orcid.org/0000-0001-5249-4151
Adam Siepel ⬤ https://orcid.org/0000-0002-3557-7219
Jerome Kelleher ⬤ https://orcid.org/0000-0002-7894-5253
Andrew D Kern ⬤ https://orcid.org/0000-0003-4381-4680

**Decision letter and Author response**
Decision letter https://doi.org/10.7554/eLife.54967.sa1
Author response https://doi.org/10.7554/eLife.54967.sa2

## Additional files
**Supplementary files**
• Transparent reporting form

**Data availability**
All resources are available from https://github.com/popsim-consortium/stdpopsim (copy archived at https://github.com/elifesciences-publications/stdpopsim).

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

## Appendix 1

### Calculating coalescence rates

In population genetics, the 'effective population size' of a population model with constant (census) size is often defined to be the number of diploids in a Wright-Fisher population that would have the same coalescence rate (or, equivalently, genetic drift) as the population in question (reviewed in *Crow and Denniston, 1988*). One reason the concept is useful is because theory predicts that genetic data from distinct populations with the same effective population size will look similar in many ways: for instance, their mean coalescence times will be the same. Conversely, this implies that effective population size should be easier to infer from genomic data than aspects of population demography that do not affect effective population size. An analogous observation holds for populations of changing size, if we define the 'coalescence rate' of a given demographic model at a particular point back in time to be the rate of coalescence of remaining lineages and define the 'coalescence effective size' at that time, denoted $N_e(t)$, so that the coalescence rate at time $t$ in the past is $1/(2N_e(t))$. With these definitions, any two models with the same effective population size trajectory ($N_e(t)$) will have the same *distribution* of coalescence times. For this reason, we might guess that if we apply an inference method that assumes a Wright-Fisher population with changing size through time to a different population model, the inferred demographic history will match the 'effective population size history' defined in this way. These observations and the following calculations are standard in coalescent theory (see e.g. *Wakeley, 2005*), but they are provided here for completeness.

We compute the coalescence rate of a collection of samples in a given demographic model at a particular point back in time as the expected number of coalescences happening at that time per unit of time and per pair of as-yet-uncoalesced lineages. More concretely, let $p(t)$ denote the probability that the lineages of a randomly chosen pair of samples have not yet coalesced $t$ units of time ago, let $p(z,t)$ denote the probability that those lineages have not yet coalesced and are furthermore both in location $z$, and let $N_e(z,t)$ be the (effective) diploid population size in location $z$ at the time, so that $1/(2N_e(z,t))$ is the rate of coalescence there. Then, we compute the mean coalescence rate as

$$r(t) = \frac{1}{p(t)} \sum_z \frac{p(z,t)}{2N_e(z,t)}.$$

This follows because if we have $m$ diploid samples, and hence $\binom{2m}{2}$ lineages, the expected number of coalescences in location $z$ between times $t$ and $t+dt$ ago is

$$\binom{2m}{2} p(z,t) \frac{dt}{2N_e(z,t)},$$

and the expected number of pairs of uncoalesced lineages at that time is

$$\binom{2m}{2} p(t).$$

The expression for $r(t)$ is a ratio of these two quantities; to obtain it we need to compute $p(t)$ and $p(z,t)$. This is relatively straightforward using the general theory of Markov chains (e.g,. *Kemeny et al., 2012*), and is implemented in `msprime`.

Note that since these quantities are *per pair of lineages*, this definition depends on the locations of the samples. The coalescence rate also has the intuitive interpretation that it is the average between-lineage coalescence rate, averaged over where uncoalesced lineages might be. Since the local coalescence rate is the inverse of the population size, $1/r(t)$ (as shown for instance in *Figure 2*) is a weighted harmonic mean of the census sizes of the different populations present at that time. This is as expected: suppose that we have two populations, one big and one small, connected by migration. If all our samples are from the big population, the number of recent coalescences should be small, reflecting the large population size, while in the long run, the coalescence rate approaches an intermediate rate. On the other hand, more recent coalescences are expected if all samples are

from the small population, A method that fits a single, time-varying population size to the data might be expected to find a population size trajectory to match these time-varying rates of coalescence.

We use the same computations to analytically compute *mean coalescence times*: since for any nonnegative random variable $T$, the mean value is $\mathbb{E}[T] = \int_0^\infty \mathbb{P}\{T{>}t\}dt$, we can obtain the mean coalescence time as

$$\int_0^\infty p(t)dt,$$

where $p(t)$ is defined above.

The coalescence rate trajectories can be computed from a model in `msprime` using the `coalescence_rate_trajectory` method of the Demography Debugger class, which can be obtained from a stdpopsim model using the `model.get_demography_debugger()` method.

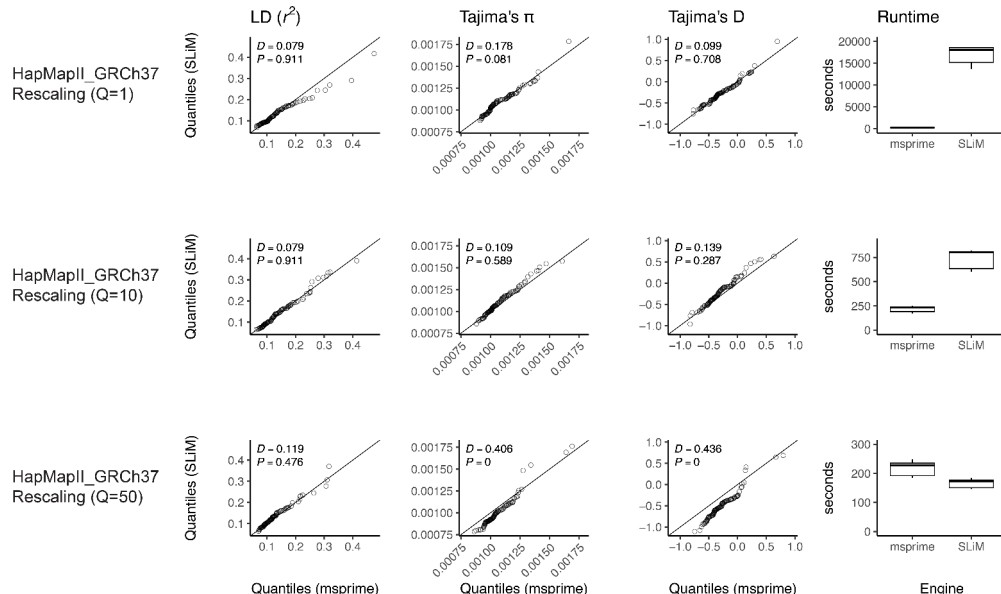

**Appendix 1—figure 1.** Validating the `SLiM` engine backend under a genetic map. Here, we validate our integration of the `SLiM` (*Haller et al., 2019*; *Haller and Messer, 2019*) engine backend. We show quantile-quantile plots between `SLiM` and `msprime` engines for three population genetic summary statistics: $r^2$, Tajima's $\pi$, and Tajima's D. Additionally, we show runtimes for generating each simulation replicate. Data were generated by simulating 100 replicates of human chromosome 22 under the AncientEurasia_9K19 model (*Kamm et al., 2019*) using the `HapMapII_GRCh37` genetic map (*Frazer et al., 2007*). 12 samples were drawn from each population (excluding basal Eurasians). From top to bottom, we show results using three scaling factors for the population sizes: Q = 1, Q = 10, and Q = 50. Kolmogorov-Smirnov two-sample test statistics (**D**) and p-values are shown, testing the null hypothesis that the quantiles were drawn from the same continuous distribution.

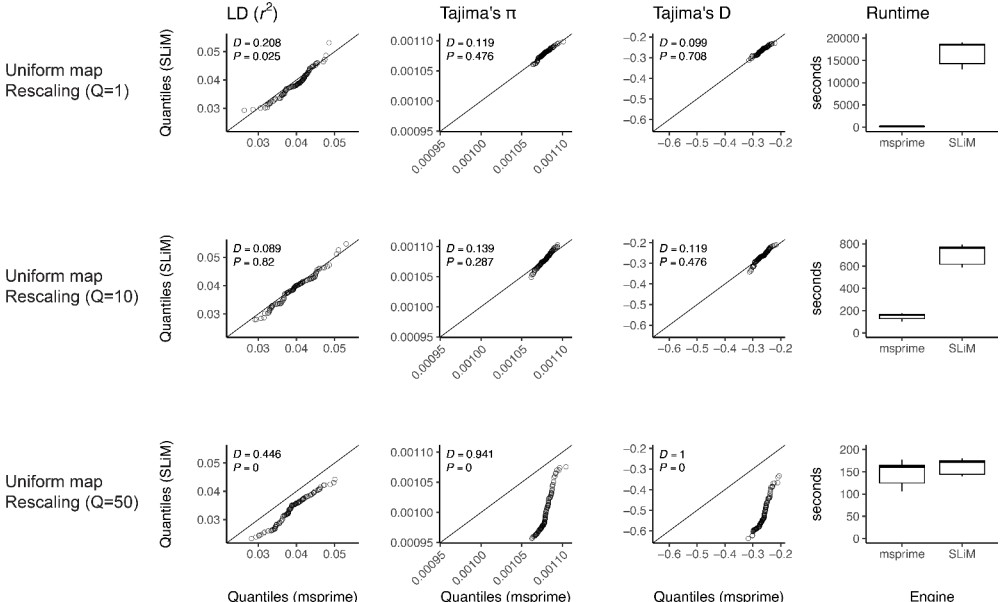

**Appendix 1—figure 2.** Validating the `SLiM` engine backend under uniform recombination. Here, we validate our integration of the `SLiM` (***Haller et al., 2019***; ***Haller and Messer, 2019***) engine backend. We show quantile-quantile plots between `SLiM` and `msprime` engines for three population genetic summary statistics: $r^2$, Tajima's $\pi$, and Tajima's D. Additionally, we show runtimes for generating each simulation replicate. Data were generated by simulating 100 replicates of human chromosome 22 under the AncientEurasia_9K19 model (***Kamm et al., 2019***) using a uniform rate of recombination across the chromosome. 12 samples were drawn from each population (excluding basal Eurasians). From top to bottom, we show results using three scaling factors for the population sizes: Q = 1, Q = 10, and Q = 50. Kolmogorov-Smirnov two-sample test statistics (**D**) and p-values are shown, testing the null hypothesis that the quantiles were drawn from the same continuous distribution.

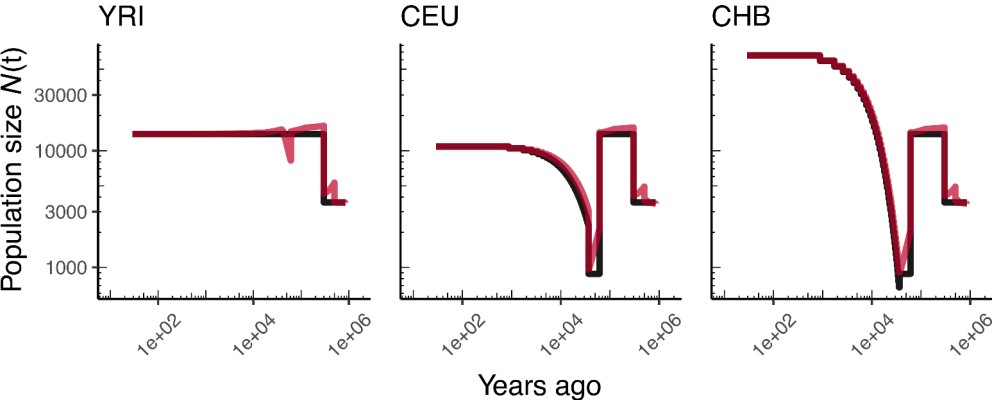

**Appendix 1—figure 3.** Comparing simulated population sizes and inverse coalescence rates in humans. Data are shown from human genomes under the `OutOfAfricaArchaicAdmixture_5R19` model (***Ragsdale and Gravel, 2019***) and using the HapMapII_GRCh37 genetic map (***Frazer et al., 2007***). From left to right, we show sizes for each of the three populations in the model: YRI, CEU, and CHB. We plot the simulated sizes for each population in black, and in red we plot inverse coalescence rates as calculated from the demographic model used for simulation (see text). In this specific model, these two measures are near identical, but in other models with higher migration rates we expect to see a larger departure between the two.

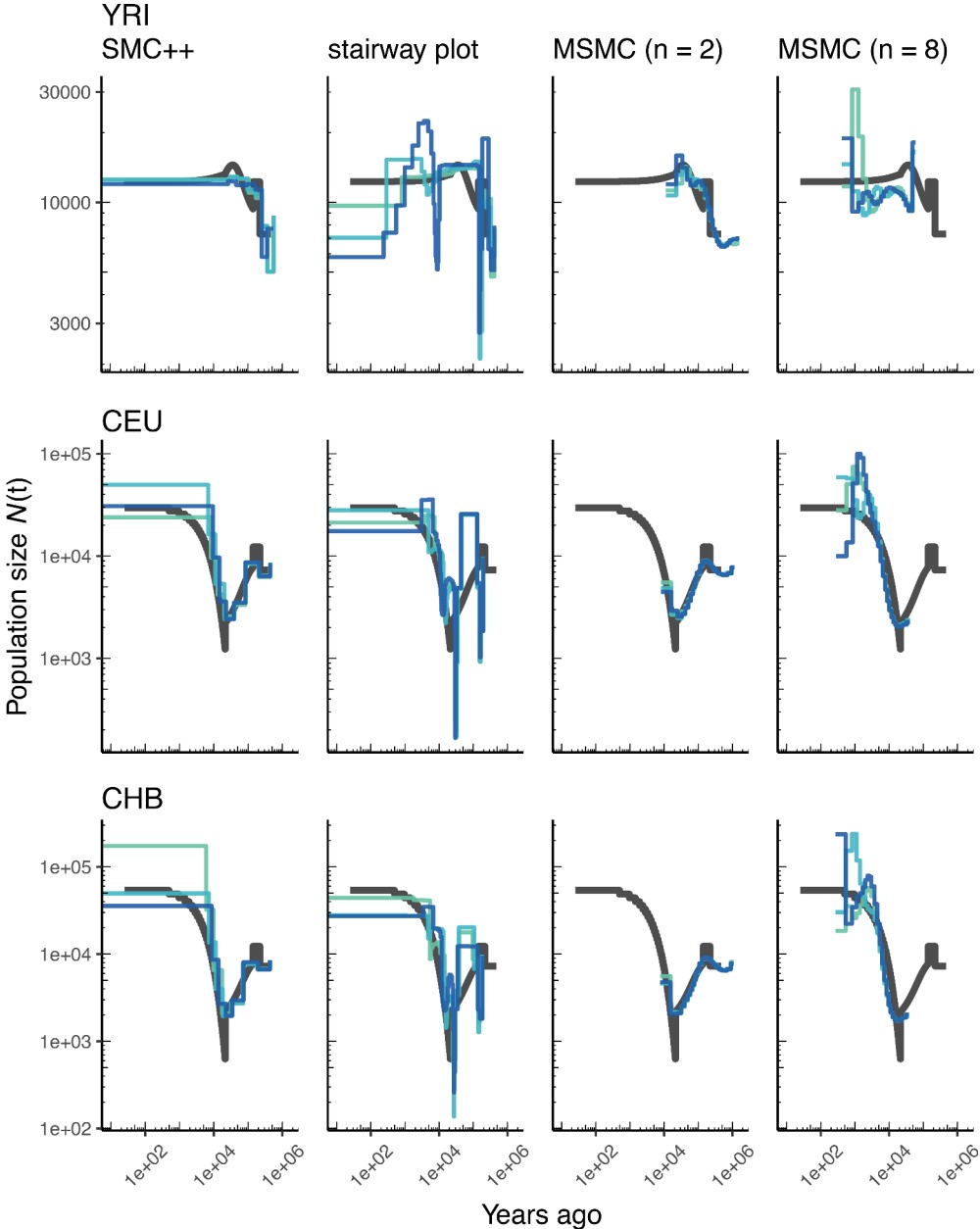

**Appendix 1—figure 4.** Comparing estimates of $N(t)$ in humans. Estimates of population size over time ($N(t)$) inferred using four different methods, `smc++`, `stairway plot`, and `MSMC` with $n = 2$ and $n = 8$. Data were generated by simulating replicate human genomes under the OutOfAfrica_3G09 model (*Gutenkunst et al., 2009*) and using the `HapMapII_GRCh37` genetic map (*Frazer et al., 2007*). From top to bottom, we show estimates for each of the three populations in the model: YRI, CEU, and CHB. In shades of blue, we show the estimated $N(t)$ trajectories for each replicate. As a proxy for the 'truth', in black we show inverse coalescence rates as calculated from the demographic model used for simulation (see text).

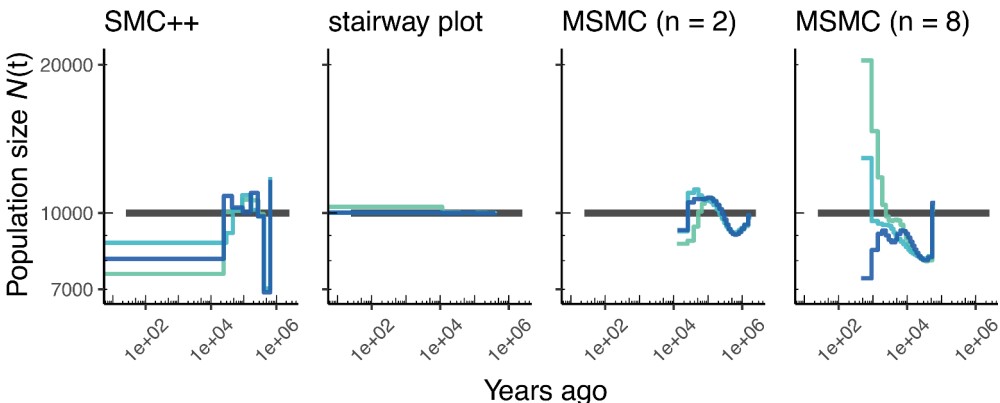

**Appendix 1—figure 5.** Comparing estimates of $N(t)$ in humans. Here, we show estimates of population size over time ($N(t)$) inferred using fourdifferent methods, `smc+`, and `stairway plot`, and `MSMC` with $n = 2$ and $n = 8$. Data were generated by simulating replicate human genomes under a constant sized population model with $N = 10^4$ and using the `HapMapII_GRCh37` genetic map (*Frazer et al., 2007*). As a proxy for the 'truth', in black we show inverse coalescence rates as calculated from the demographic model used for simulation (see text).

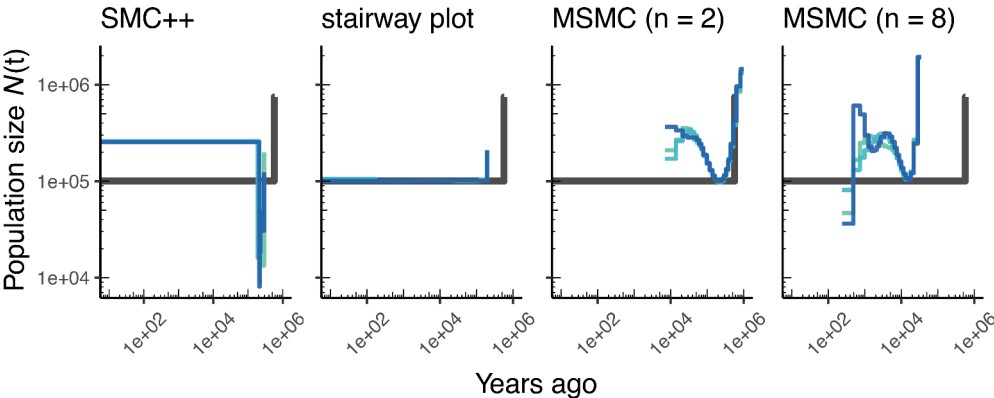

**Appendix 1—figure 6.** Comparing estimates of $N(t)$ in *A. thaliana*. Here, we show estimates of population size over time ($N(t)$) inferred using four different methods, `smc++`, and `stairway plot`, and `MSMC` with $n = 2$ and $n = 8$. Data were generated by simulating replicate *A. thaliana* genomes under the `African2Epoch_1H18` model (*Durvasula et al., 2017*) and using the `SalomeAveraged_TAIR7` genetic map (*Salomé et al., 2012*). As a proxy for the 'truth', in black we show inverse coalescence rates as calculated from the demographic model used for simulation (see text).

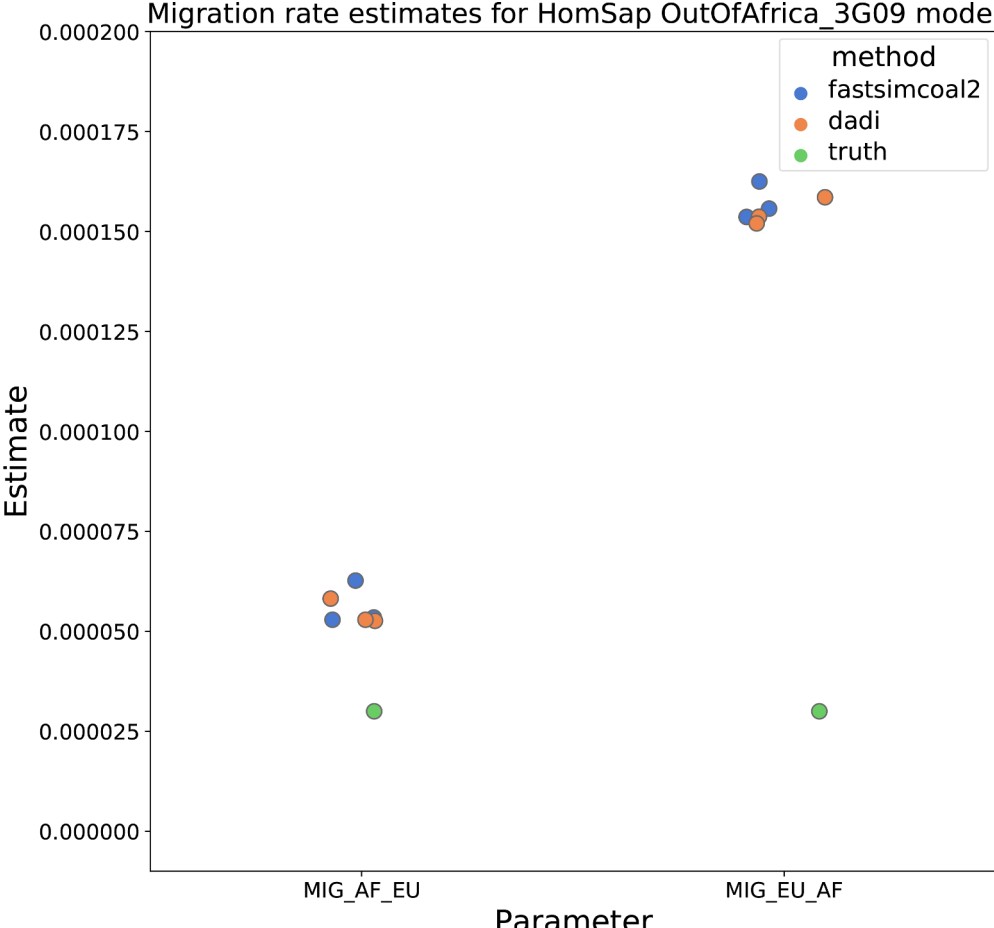

**Appendix 1—figure 7.** Migration rate estimates for the human Gutenkunst model. Here, we show inferred migration rates from ∂*a*∂*i* and `fastsimcoal2`. Data were generated by simulating replicate human genomes under the *Gutenkunst et al., 2009* model and using the genetic map inferred in *Frazer et al., 2007*. Directional migration from Europe to Africa is represented as *MIG_AF_EU* and migration from Africa to Europe is represented as *MIG_EU_AF*. Note that the *x*-axis coordinates are arbitrary.

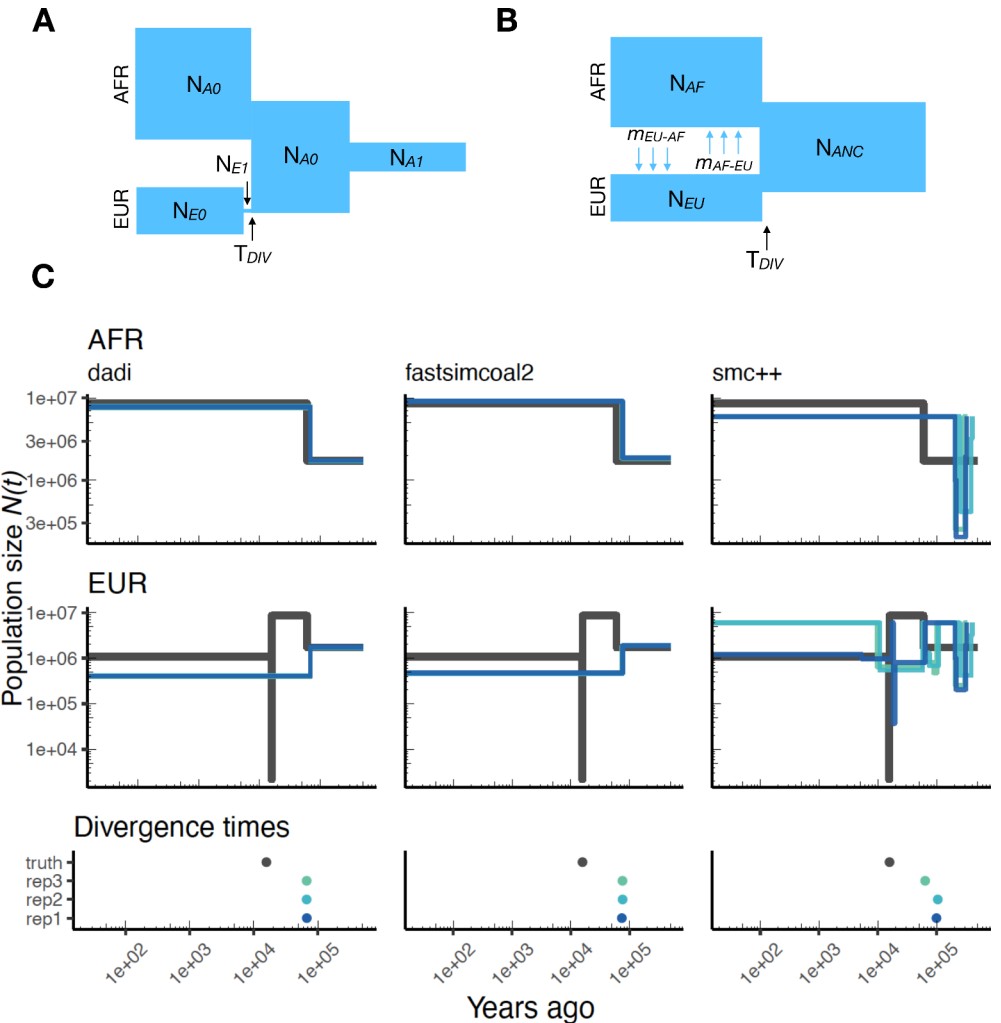

**Appendix 1—figure 8.** Parameters estimated using a two-population *Drosophila* model. Here, we show estimates of $N(t)$ inferred using ∂a∂i, `fastsimcoal2`, and `smc++`. Data were generated by simulating replicate *Drosophila* genomes under the *Li and Stephan, 2006* model and using the genetic map inferred in *Comeron et al., 2012*. See legend of *Figure 4* for details. In shades of blue, we show the estimated $N(t)$ trajectories for each replicate. In black we show the simulated population sizes.

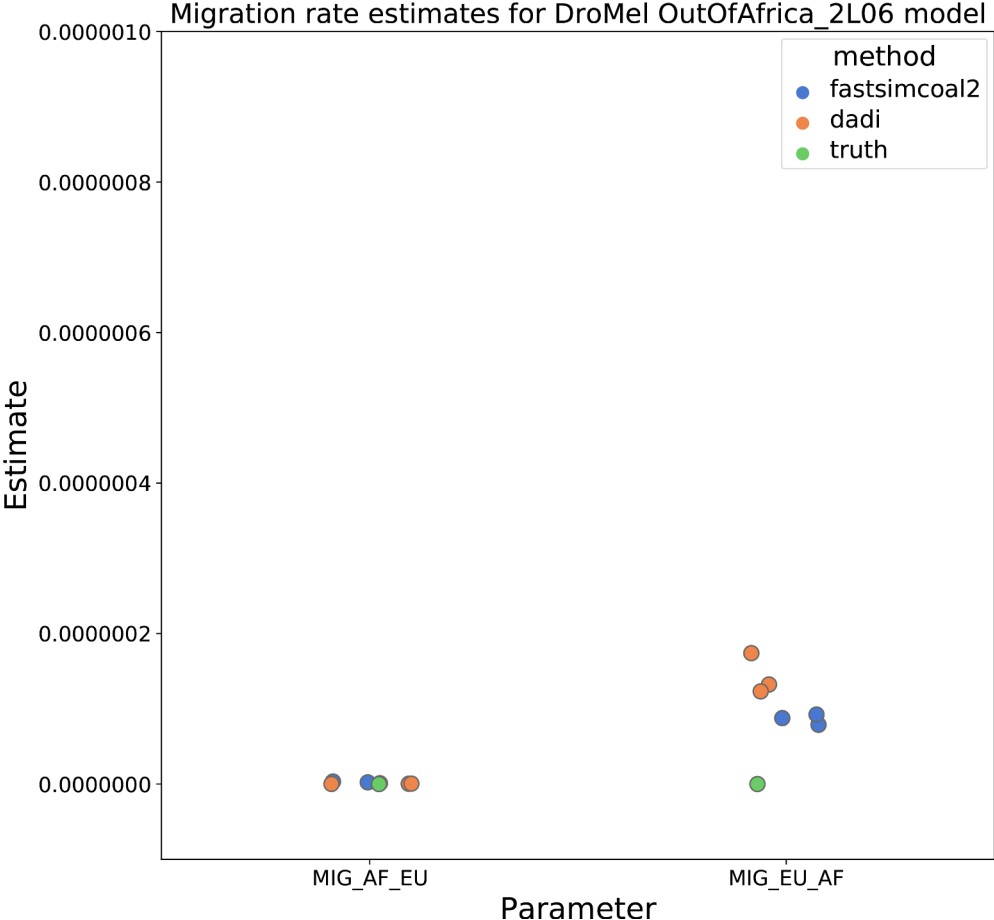

**Appendix 1—figure 9.** Migration rate parameters estimated under a two-population *Drosophila* model. Here, we show inferred migration rates from $\partial a \partial i$ and `fastsimcoal2`. Data were generated by simulating replicate *Drosophila* genomes under the *Li and Stephan, 2006* model and using the genetic map inferred in *Comeron et al., 2012*. Directional migration from Europe to Africa is represented as $MIG\_AF\_EU$ and migration from Africa to Europe is represented as $MIG\_EU\_AF$. Note that the $x$-axis coordinates are arbitrary.

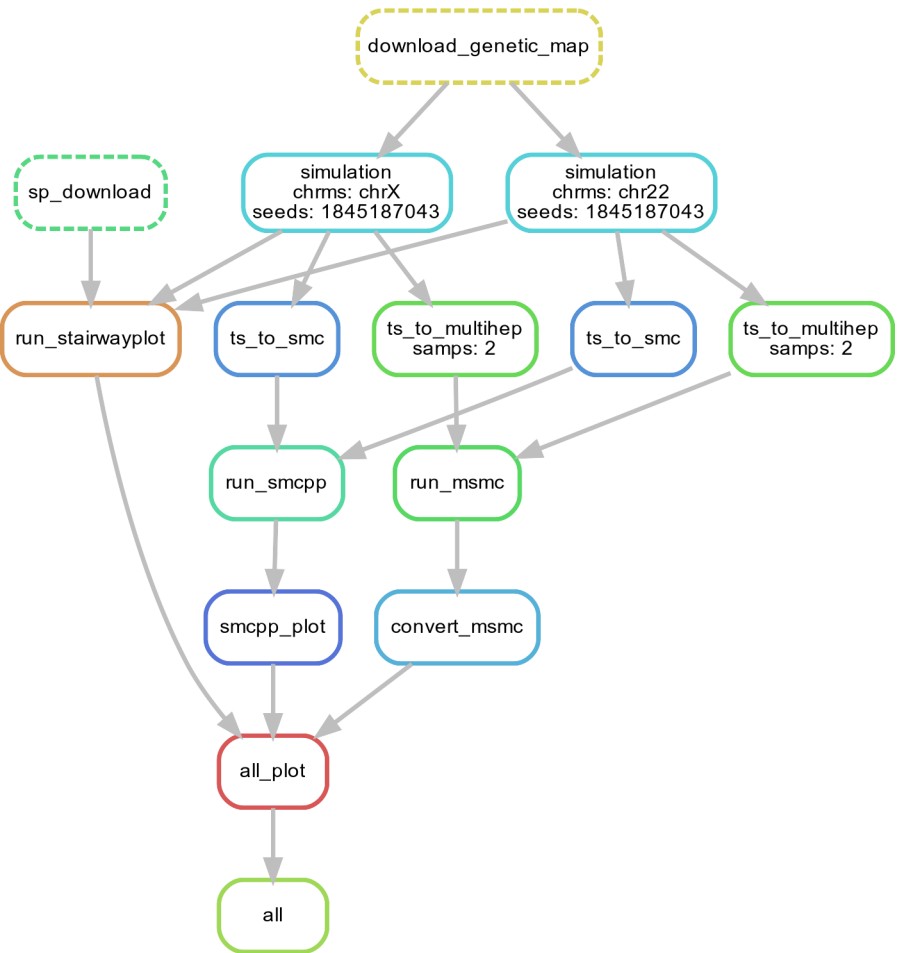

**Appendix 1—figure 10.** Workflow for our N(t) inference methods comparison. Here, we show single replicate for two chromosomes, chr22 and chrX, simulated under the HomSap OutOfAfrica_3G09 demographic model, with a HapmapII_GRCh37 genetic map. Note that the data used as input by all inference methods `smc++`, MSMC, and `stairway plot`, come from the same set of simulations.

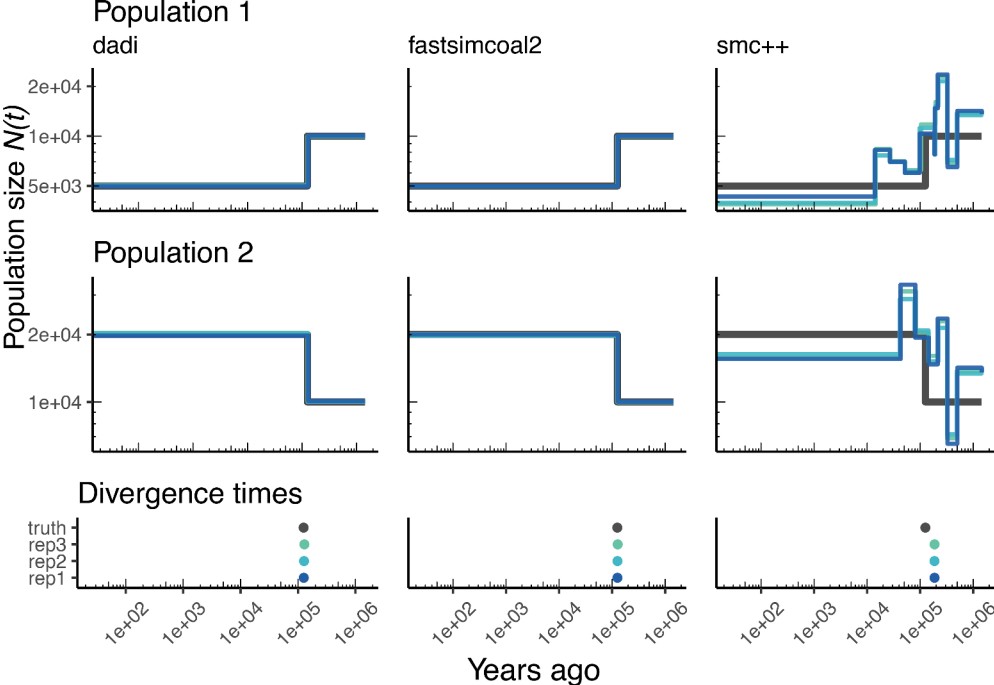

**Appendix 1—figure 11.** Parameters estimated from a generic IM model Here we show estimates of $N(t)$ inferred using ∂a∂i, `fastsimcoal2`, and `smc++`. Data were generated by simulating under a generic IM model with a human genome and *Frazer et al., 2007* genetic map. In shades of blue we show the estimated $N(t)$ trajectories for each replicate. In black we show the simulated population sizes.

