## [Decision Letter]

**Acceptance summary:**

Simulations have long been central to population genetics. Population genomics has, in turn, become central to numerous areas of evolutionary biology and human genetics, and a vast array of different statistical methods have been developed. However, these methods are rarely ground-truthed in any truly reproducible manner. This paper takes an important set of steps in developing a flexible framework to standardize testing of population genetics software.

**Decision letter after peer review:**

Thank you for submitting your article "A community-maintained standard library of population genetic models" for consideration by *eLife*. Your article has been reviewed by three peer reviewers, and the evaluation has been overseen by a Reviewing Editor and Patricia Wittkopp as the Senior Editor. The following individuals involved in review of your submission have agreed to reveal their identity: John Novembre (Reviewer #1); Arun Sethuraman (Reviewer #2); Sara Mathieson (Reviewer #3).

The reviewers have discussed the reviews with one another and the Reviewing Editor has drafted this decision to help you prepare a revised submission.

Simulations have long been central to population genetics. Population genomics has, in turn, become central to numerous areas of evolutionary biology and human genetics, and a vast array of different statistical methods have been developed. However, these methods are rarely ground-truthed in any truly reproducible manner. This paper takes an important set of steps in developing a flexible framework to standardize testing of population genetics software.

The reviewers' comments are all reasonably straight forward, often hit on similar points, and so most should be easily addressable. Please respond to each point in your response to reviews. In our online discussions with the reviewers three points came the fore:

1) Ensuring that the language about the models and parameters conveys the correct sense of uncertainty. Obviously the reviewers and the authors know that these just represent best guesses at the moment, but as this platform catches on we don't want these numbers and models to be seen as gospel.

2) We would like to see the SLIM integration demonstrated with an application that MsPrime doesn't cover. This could be, for example, a figure of the average value some summary statistics surrounding a selective under the human demographic model. The reviewers didn't want this to be a lot of work, e.g. it would be outside of the scope of the current paper to demonstrate the power of a range of selection methods. However, we felt that a simple demonstration of the functionality would substantially increase the scenarios easily available and imaginable by readers of the paper.

3) We felt that the Discussion would benefit from a discussion of the details not yet incorporated into the platform, e.g. sex-specific recombination, gene conversion, assembly error, structural variants etc. Such an addition might spur future development, and also give the authors an opportunity to discuss general ways forward on these issues.

4) We would be interested to see more of a discussion of the strengths and weakness of the various demographic inference methods.

Beyond testing methods, another major use axis that I can personally see for this platform is for empirical researchers to see whether patterns they see in their data are consistent with known demographic models. For example, demonstrating whether the frequency spectrum in larger samples lines up with previously inferred models. Having all of these models implemented in a central place will significantly lower the barrier of entry of empirical researchers into rigorous simulation frameworks.

Overall we are very excited to see this big move forward and welcome the authors' careful work in this important area.

Reviewer #1:

The manuscript describes a community effort to standardize population genetic simulations and it presents an example of the resource's utility for method's testing. This development answers a long-standing need in the population genetics community for greater standardization and ease-of-implementation of simulation protocols. As such, I expect it will be a well-used resource and it represents an important advance for the field's practices. From the test implementation included in the paper, it was remarkable to see how highly variable (and often poor) the performance of the demographic inference methods was, and those results add to the scientific contribution of this manuscript.

Some of the specific features that are nice about the work are the incorporation of inferred genetic maps where possible, the automated output of citation information, the use of long-term stable identifiers for models and genetic maps, and the outputting of provenance information in the tree sequence files.

1) Emphasize parameters are current best-guesses:

a) I am most worried about the fragility of the models over time and the misperception one might have that these models are "accurate" for a species. Genome assemblies change, recombination rates improve, mutation rates change, etc. I would like to see that uncertainty reflected more in the language used in the paper so that it's clear the catalog is a collection of inferred parameters that are subject to change over time. This is a subtle and cosmetic, but important I think. For example, "the library defines basic information about each species genome, including information about chromosome lengths, average mutation rates, and generation times." Yet, this is information we don't have – each of these parameters is actively under revision/discussion for even the best studied species. It would be great to hammer home these are all inferred values. (Also see sentence on "details on the physical organization of its genome"; the same comment applies, and an improvement might be something like "details on the physical organization of the latest reference genome assembled for the species").

b) On a related note, there is wording regarding ensuring "implemented models are accurate" – I think what is meant is that the "implemented models are faithful to the source publications from which they derive". As the second half of this paper makes clear, because of errors in inference, many of these models will not be accurate, in the sense of representing the true history of the species.

2) It would be wonderful to have a comments section for the models that would be either curated by a set of editors or crowd-sourced. I say this because overtime, models will proliferate, and some will come to be regarded as out-of-date. One can imagine those approaching the field afresh will be overwhelmed by the possible selections and possible implement models that become outdated. If the goal is standardization, how do we communicate that standards change? A comment system (or even star-rating system?) may be wise to implement now. Assuming it can be layered on top of the existing structure, it may be enough for this publication to note this as a future challenge that needs development/addressing.

3) In terms of the maturity of the examples developed for the initial release, I would have liked to see at least one simulation model with a selective sweep, one with background selection, and one with spatial stepping-stone structure. Each of these would be helpful test cases to implement to be sure that the existing catalog framework has the breadth/flexibility necessary to accommodate future use cases. I do not think this is a requirement for publication, but it would add great value to this initial release of the resource.

4) The approach of masking "low-recombination" portions of the chromosomes seems like an incomplete/indirect attempt to model the inherent limitations of sequencing to an "accessible" genome.

a) Shouldn't the approach instead be to drop "low complexity" regions (e.g., as defined by an excessive number of "N"'s in the reference, low mapability scores, or via tools like RepeatMasker?). This part of the pipeline seems open to refinement.

b) Are the "masks" a separate configuration file for the simulations? It seems that it would be preferable for them to be separate from the recombination rate files – right now it reads as if the mask applied is a function of the genetic map file, but this seems too inflexible for users who prefer an alternative approach to masking.

Reviewer #2:

The authors in this manuscript describe the implementation of a publicly curated, open source simulation package called stdpopsim – equipped with commonly utilized population genetic demographic models in humans, *Drosophila melanogaster*, and *Arabidopsis thaliana*. I am in awe of what these authors have achieved, in terms of benchmarking these standard models in an effort to avoid duplication of effort, and possibility of erroneous inference. The package is currently equipped with several "in-built" models that allow the simulation of trees with msprime and SLiM. The authors explain the application of these models by simulating and benchmarking estimates of Ne under a couple of scenarios. The manuscript is also well written, and use popular tools like dadi and smc++ to estimate and benchmark the simulations under a variety of models. Across all simulated scenarios (except under more complex models), the simulations seem relatively accurate.

Despite a little hiccup with python version mismatch prior to me successfully installing stdpopsim, I was able to successfully get the tutorials running within minutes after. I have one recommendation however for the tutorials – it would definitely help if the CLI versus python tutorials were kept separate. I found it a little confusing since they are all listed on the same page (https://stdpopsim.readthedocs.io/en/latest/tutorial.html). The simulations, testing models, calculating divergences, plotting ran without a hitch, and I am impressed and excited to play around with more models in coming days. Having also developed similar libraries/pipelines, I have also found it extremely useful for developers to provide some more detailed documentation/tutorials via Jupyter notebooks, or some similar platform. I did however notice that the authors have provided their analyses as Snakemake files in the interest of replication. I did not replicate their analyses, but I trust that the documentation for these analyses are detailed enough to aid readers/users in establishing similar analysis pipelines for stdpopsim simulated data.

I thoroughly enjoyed reading this manuscript, and learning of all the new features that have been written into stdpopsim, and believe that this will be an invaluable contribution to the population genomics community.

Reviewer #3:

The authors present a standardized framework for creating reproducible population genetic simulations. This resource will allow researchers to create models for new species/scenarios, and easily compare methods on the same dataset. The authors are correct that the current state of benchmarking in population genetics causes inconsistency, duplicated effort, and confusion about the performance of different methods. The authors highlight that creating a realistic and meaningful simulation study is a barrier to entering this field, and I absolutely agree. I will be passing this resource on to my students and look forward to using it myself. Stdpopsim is a crucial step forward. The manuscript itself is well-written and describes an extensive comparison of demography methods in a variety of species/scenarios. I have a few comments.

1) The authors mention that SLiM can be used as an alternative backend, which would presumably allow for simulations with selection. Although I don't think an extensive comparison of selection methods is necessary for this paper, it would be ideal if the authors can give some idea of how this would work (example command line, etc). There are also a myriad of methods for detecting/quantifying selection, and these simulations are not consistent either.

2) I like the inclusion of the "zigzag" history, as well as generic piecewise constant models and IM models (subsection “The Species Catalog”). I wonder if these could be included in a separate section (not organism specific) in the documentation and software (and then in Table 1). Right now the zigzag model is under humans in the catalog.

3) In the subsection “Use case: comparing methods of demographic inference”, the authors set up notation for the number of replicates (R), number of chromosomes (C), and sample size (n), but don't seem to use it afterward (or use it inconsistently). It would be helpful if all the figure legends and main text included this notation (I am guessing the number of replicates is 3 based on the images, but this should be clarified). The authors use N in the Materials and methods (i.e. subsection “Workflow for analysis of simulated data”) to refer to population size (which makes sense), but then also say "In all cases we set the sample size of the focal population to N = 50 chromosomes." For MSMC, the sample size was set to n=2,8 which suggests haploid samples, but the "Calculating coalescence rates" section says that n is the diploid sample size.

4) "Calculating coalescence rates" section needs a read through. Reword first sentence and add some citations (especially regarding computing p(t) and p(z,t)). It was unclear to me how the "mean coalescence times" were used (the rate was used to compute the ground truth over time). This section is also referred to as the Appendix in the main text.

---

## [Author Response]

[…] In our online discussions with the reviewers three points came the fore:1) Ensuring that the language about the models and parameters conveys the correct sense of uncertainty. Obviously the reviewers and the authors know that these just represent best guesses at the moment, but as this platform catches on we don't want these numbers and models to be seen as gospel.

We fully agree that the language in the original manuscript was not careful enough in conveying uncertainty about the models and parameters we discuss. We have made edited the text throughout the manuscript to address this. See our response to reviewer 1 below for examples. We have also added a bit more to the Discussion (below “next steps”) on this point.

2) We would like to see the SLIM integration demonstrated with an application that MsPrime doesn't cover. This could be, for example, a figure of the average value some summary statistics surrounding a selective under the human demographic model. The reviewers didn't want this to be a lot of work, e.g. it would be outside of the scope of the current paper to demonstrate the power of a range of selection methods. However, we felt that a simple demonstration of the functionality would substantially increase the scenarios easily available and imaginable by readers of the paper.

We would like to see this also and agree that it would be of great interest. However, we felt that performing adequate simulations with selection would require a substantial amount of additional work and would significantly expand the scope of the manuscript. Furthermore, discussing selection would distract from the current relatively simple focus on inference of demographic history. We definitely intend to include selection (and expand the available generic models) in future work, as we now discuss in greater detail in the Discussion (“next steps”).

Instead, we have decided to add a section validating the use of SLiM in stdpopsim neutral simulations (see “simulation engines” section in the Results and Appendix—figures 1 and 2). Our focus there was to show that under neutral simulations, SLiM produced data that were consistent with the coalescent simulations of msprime. To expand on this fundamental point, we also examined the influence of population size down-scaling, which is a common practice used in forward simulations, and indeed often crucial for tractable compute times. While this is by no means a comprehensive investigation into this issue, we do believe that it demonstrates the use of SLiM in future applications that will also simulate selection.

3) We felt that the Discussion would benefit from a discussion of the details not yet incorporated into the platform, e.g. sex-specific recombination, gene conversion, assembly error, structural variants etc. Such an addition might spur future development, and also give the authors an opportunity to discuss general ways forward on these issues.

Thank you for this good idea – we have expanded the section of the Discussion

(“next steps”) with more detail on our future plans. While that is so we are wary of promising too much at this point.

4) We would be interested to see more of a discussion of the strengths and weakness of the various demographic inference methods.

We have added an additional paragraph to the Discussion exploring various factors that may affect choice of inference method. These include the data required by the method, the type of model, and the implementation. We also provide a qualitative comparison between the performance of methods, as shown in our limited analysis. We believe that a more comprehensive comparison between methods is beyond the scope of the current manuscript, which focuses on the resource itself, rather than its application.

Beyond testing methods, another major use axis that I can personally see for this platform is for empirical researchers to see whether patterns they see in their data are consistent with known demographic models. For example, demonstrating whether the frequency spectrum in larger samples lines up with previously inferred models. Having all of these models implemented in a central place will significantly lower the barrier of entry of empirical researchers into rigorous simulation frameworks.

This is an important point – we have (naturally) focused on methods benchmarking, but our impact could well be greater if stdpopsim becomes widely used among empirical researchers (for whom on average running realistic simulations presents a greater barrier than to developers of computational methods). We’ve added a paragraph about this to the Discussion and additional words to the Abstract.

Reviewer #1:[…] 1) Emphasize parameters are current best-guesses:a) I am most worried about the fragility of the models over time and the misperception one might have that these models are "accurate" for a species. Genome assemblies change, recombination rates improve, mutation rates change, etc. I would like to see that uncertainty reflected more in the language used in the paper so that it's clear the catalog is a collection of inferred parameters that are subject to change over time. This is a subtle and cosmetic, but important I think. For example, "the library defines basic information about each species genome, including information about chromosome lengths, average mutation rates, and generation times." Yet, this is information we don't have – each of these parameters is actively under revision/discussion for even the best studied species. It would be great to hammer home these are all inferred values. (Also see sentence on "details on the physical organization of its genome"; the same comment applies, and an improvement might be something like "details on the physical organization of the latest reference genome assembled for the species").

We thank the reviewer for bringing up this important point and fully agree that the language in the manuscript needs to better emphasize the uncertainty in our current parameter estimates. To address this, we have made several changes throughout the manuscript, including the passages cited above:

The first quoted passage now reads: “For each species, the catalog contains curated information on our current understanding of the physical organization of its genome, inferred genetic maps, population-level parameters (e.g., mutation rate and generation time estimates), and published demographic models. These models and parameters are meant to represent the field’s current understanding, and we intend for this resource to evolve as new results become available, and other existing models are added to stdpopsim by the community.”

The second passage now reads: “Firstly, the library defines some basic information about our current understanding of each species’ genome, including information about chromosome lengths, average mutation rate estimates, and generation times. We also provide access to detailed empirical information such as inferred genetic maps, which model observed heterogeneity in recombination rate along chromosomes.”

b) On a related note, there is wording regarding ensuring "implemented models are accurate" – I think what is meant is that the "implemented models are faithful to the source publications from which they derive". As the second half of this paper makes clear, because of errors in inference, many of these models will not be accurate, in the sense of representing the true history of the species.

We made the suggested change: “Importantly, we developed rigorous quality control methods to ensure that we have correctly implemented the models as described in their original publication and provided documented methods for others to contribute new models.”

2) It would be wonderful to have a comments section for the models that would be either curated by a set of editors or crowd-sourced. I say this because overtime, models will proliferate, and some will come to be regarded as out-of-date. One can imagine those approaching the field afresh will be overwhelmed by the possible selections and possible implement models that become outdated. If the goal is standardization, how do we communicate that standards change? A comment system (or even star-rating system?) may be wise to implement now. Assuming it can be layered on top of the existing structure, it may be enough for this publication to note this as a future challenge that needs development/addressing.

Thank you for this forward-thinking comment. We considered various ways to do this (e.g., by enabling the wiki associated with the github repository: https://github.com/popsim-consortium/stdpopsim/issues/415)

3) In terms of the maturity of the examples developed for the initial release, I would have liked to see at least one simulation model with a selective sweep, one with background selection, and one with spatial stepping-stone structure. Each of these would be helpful test cases to implement to be sure that the existing catalog framework has the breadth/flexibility necessary to accommodate future use cases. I do not think this is a requirement for publication, but it would add great value to this initial release of the resource.

As mentioned in our response to the editor’s comments, we agree that this is an important avenue to pursue in the near future. However, performing adequate simulations with selection would require a substantial amount of additional work and would likely distract from the current relatively simple focus on inference of demographic history. We definitely intend to include selection (and expand the available generic models) in future work, as we now discuss in greater detail in the Discussion (“next steps”).

4) The approach of masking "low-recombination" portions of the chromosomes seems like an incomplete/indirect attempt to model the inherent limitations of sequencing to an "accessible" genome.a) Shouldn't the approach instead be to drop "low complexity" regions (e.g., as defined by an excessive number of "N"'s in the reference, low mapability scores, or via tools like RepeatMasker?). This part of the pipeline seems open to refinement.

Our initial motivation for masking was to reduce the overrepresentation of marginal trees with little to no recombination from biasing patterns of diversity in such a way that demographic inference methods would be misled. While we agree with the reviewer that different masking approaches might better reflect the masking done on real genomic data, at this time the best way to mask remains an open question, and we feel that a nuanced analysis of masking is outside the scope of this paper. For these reasons, we feel that masking using a simple recombination rate threshold is a reasonable approach. We note that there is however a substantial correlation between recombination rate and mappability (by any of the measures mentioned above) wherein the notoriously difficult portions of genomes to assemble are generally in regions of low recombination. We agree that there may indeed be better ways to mask, which is why we allow users to mask their simulations as they see fit.

b) Are the "masks" a separate configuration file for the simulations? It seems that it would be preferable for them to be separate from the recombination rate files – right now it reads as if the mask applied is a function of the genetic map file, but this seems too inflexible for users who prefer an alternative approach to masking.

Mask files are not currently a component of stdpopsim proper, rather they were implemented separately from running stdpopsim, for the sole purpose of comparing demographic inference methods. All of the masks that we have used are available on the analysis repository that is associated with this manuscript. Users who download stdpopsim will always be simulating raw and unmasked tree sequence files, to which they can apply any variety of masks post hoc, if they so choose. We are currently making plans, however, to incorporate some version of this process into a future stdpopsim release.

Reviewer #2:[…] Despite a little hiccup with python version mismatch prior to me successfully installing stdpopsim, I was able to successfully get the tutorials running within minutes after. I have one recommendation however for the tutorials – it would definitely help if the CLI versus python tutorials were kept separate. I found it a little confusing since they are all listed on the same page (https://stdpopsim.readthedocs.io/en/latest/tutorial.html). The simulations, testing models, calculating divergences, plotting ran without a hitch, and I am impressed and excited to play around with more models in coming days. Having also developed similar libraries/pipelines, I have also found it extremely useful for developers to provide some more detailed documentation/tutorials via Jupyter notebooks, or some similar platform. I did however notice that the authors have provided their analyses as Snakemake files in the interest of replication. I did not replicate their analyses, but I trust that the documentation for these analyses are detailed enough to aid readers/users in establishing similar analysis pipelines for stdpopsim simulated data.

Good idea – we have reorganized the Tutorials and added to them, and have also added in a basic stdpopsim API and CLI example in a Jupyter Notebook that can be accessed and used interactively via Binder and encourage users to try out the tutorials there. We have linked to the Binder in the README on the GitHub. See https://mybinder.org/v2/gh/popsimconsortium/stdpopsim/master?filepath=stdpopsim_example.ipynb

Reviewer #3:[…] I have a few comments.1) The authors mention that SLiM can be used as an alternative backend, which would presumably allow for simulations with selection. Although I don't think an extensive comparison of selection methods is necessary for this paper, it would be ideal if the authors can give some idea of how this would work (example command line, etc). There are also a myriad of methods for detecting/quantifying selection, and these simulations are not consistent either.

As mentioned in our response to the editor’s comment, we added a demonstration of the use of SLiM to the Results (see “Simulation engines” section and Appendix—figures 2 and 2). See also our response to related comments made by reviewers #1 and #2.

2) I like the inclusion of the "zigzag" history, as well as generic piecewise constant models and IM models (subsection “The Species Catalog”). I wonder if these could be included in a separate section (not organism specific) in the documentation and software (and then in Table 1). Right now the zigzag model is under humans in the catalog.

We considered both of these suggestions, and here’s why we’ve left it the way it is. Most of the columns of the table don’t apply to generic models, so it seems strange to include them there. The “zigzag” model could definitely be a generic model, and indeed was initially implemented as such. However, the reason we settled on the zigzag model is being defined as a human model (see discussion at https://github.com/popsim-consortium/stdpopsim/issues/106) is that its effective population size values are taken from (or at least inspired by) values inferred from human genomes: as implemented, it is not as “generic” as it might seem.

3) In the subsection “Use case: comparing methods of demographic inference”, the authors set up notation for the number of replicates (R), number of chromosomes (C), and sample size (n), but don't seem to use it afterward (or use it inconsistently). It would be helpful if all the figure legends and main text included this notation (I am guessing the number of replicates is 3 based on the images, but this should be clarified). The authors use N in the Materials and methods (i.e. subsection “Workflow for analysis of simulated data”) to refer to population size (which makes sense), but then also say "In all cases we set the sample size of the focal population to N = 50 chromosomes." For MSMC, the sample size was set to n=2,8 which suggests haploid samples, but the "Calculating coalescence rates" section says that n is the diploid sample size.

This is a good catch! We see why this is confusing. We introduced this notation

(R, C, and n) to make it completely clear in this paragraph what exactly was being simulated, but we feel that continuing to use this notation elsewhere would actually obscure things, since we don’t do any calculations with these quantities. We have changed the “n” in the Appendix to an “m”.

4) "Calculating coalescence rates" section needs a read through. Reword first sentence and add some citations (especially regarding computing p(t) and p(z,t)). It was unclear to me how the "mean coalescence times" were used (the rate was used to compute the ground truth over time). This section is also referred to as the Appendix in the main text.

We’ve expanded the first sentence substantially, and added a few more citations. We apologize that we don’t know of a paper to cite that does precisely the same calculations (but don’t doubt that such a paper exists), and instead refer only vaguely to “general theory of Markov chains” (but now with a citation). The section is now explicitly labeled “Appendix”. Hopefully this is more clear now.